



# Warm conveyor belts in present-day and future climate simulations. Part II: Role of potential vorticity production for cyclone intensification

Hanin Binder[1], Hanna Joos[1], Michael Sprenger[1], and Heini Wernli[1]

[1]Institute for Atmospheric and Climate Science, ETH Zurich, 8092 Zurich, Switzerland

**Correspondence:** Hanin Binder (hanin.binder@env.ethz.ch)

**Abstract.**

Warm conveyor belts (WCBs) are strongly ascending, cloud and precipitation forming airstreams in extratropical cyclones. The intense cloud-diabatic processes produce low-level cyclonic potential vorticity (PV) along the ascending airstreams, which often contribute to the intensification of the associated cyclone. This study investigates how climate change affects the cyclones' WCB strength and the importance of WCB-related diabatic PV production for cyclone intensification, based on present-day (1990-1999) and future (2091-2100) climate simulations of the Community Earth System Model Large Ensemble (CESM-LE). In each period, a large number of cyclones and their associated WCB trajectories have been identified in both hemispheres during the winter season. Compared to ERA-Interim reanalyses, the present-day climate simulations are able to capture the cyclone structure and the associated WCBs remarkably well, which gives confidence in future projections with CESM-LE. The comparison of the simulations reveals an increase in the WCB strength and the cyclone intensification rate in the Southern Hemisphere (SH) in the future climate. The WCB strength also increases in the Northern Hemisphere (NH), but to a smaller degree, and the cyclone intensification rate is not projected to change considerably. Hence, in the two hemispheres cyclone intensification responds differently to an increase in WCB strength, which however is consistent with the opposite changes in near-surface baroclinicity expected with rising temperatures. Indeed, baroclinicity is expected to increase in the SH, which interacts positively with the direct effects of enhanced WCB-related diabatic heating to produce stronger cyclones, whereas it is expected to decrease in the NH, which counteracts the effects of the moist processes in the WCB ascent. Cyclone deepening correlates positively with the intensity of the associated WCB, with a Spearman correlation coefficient of 0.68 (0.66) in the NH in the present-day (future) simulations, and a coefficient of 0.51 (0.55) in the SH. The number of explosive cyclones with strong WCBs, referred to as C1 cyclones, is projected to increase in both hemispheres, while the number of explosive cyclones with weak WCBs (C3 cyclones) is projected to decrease. A composite analysis reveals that in the future climate C1 cyclones will be associated with even stronger WCBs, more WCB-related diabatic PV production, the formation of a more intense PV tower, and an increase in precipitation. They will become warmer, moister, and slightly more intense. The findings indicate that (i) cyclones will be more diabatic in a warmer climate, (ii) WCB-related PV production will be even more important for explosive cyclone intensification than in the present-day climate, and (iii) the interplay between dry and moist dynamics is crucial to understand how climate change affects cyclone intensification.





## 1  Introduction

In extratropical cyclones, most of the cloud and precipitation formation occurs in so-called warm conveyor belts (WCBs; e.g., Harrold, 1973; Browning, 1990; Wernli and Davies, 1997). WCBs are coherent airstreams that originate in the cyclone's warm sector and rapidly move poleward while ascending to the upper troposphere. They form elongated cloud bands with warm clouds in their inflow, mixed-phase clouds at mid-levels and ice clouds in the upper troposphere (Joos and Wernli, 2012; Wernli et al., 2016; Binder et al., 2020). The intense cloud-diabatic processes within the ascending airstreams modify potential vorticity (PV) in the atmosphere (Wernli and Davies, 1997; Madonna et al., 2014). In the WCB outflow in the upper troposphere, diabatic PV reduction results in negative PV anomalies, which can influence the synoptic- and large-scale flow (Wernli, 1997; Pomroy and Thorpe, 2000; Grams et al., 2011) and contribute to the formation and amplification of forecast errors (e.g., Madonna et al., 2015; Martínez-Alvarado et al., 2016; Grams et al., 2018; Binder et al., 2021). In the lower troposphere, diabatic PV production generates positive PV anomalies that can contribute to the intensification of the associated cyclone (Davis and Emanuel, 1991; Stoelinga, 1996; Binder et al., 2016). In particular, diabatic PV production within the WCB is crucial for many explosively intensifying cyclones (Binder et al., 2016). So far, it is unclear how global warming affects the cyclones' WCB strength and the importance of WCB-related diabatic PV production for cyclone intensification.

The response of extratropical cyclones to global warming is uncertain and a topic of ongoing research (see, e.g., the reviews by Ulbrich et al., 2009; Shaw et al., 2016; Catto et al., 2019). Multiple, partly opposing mechanisms are considered to be affecting the intensity, frequency and geographical distribution of cyclones in a future climate: (i) the atmospheric moisture content is expected to increase in a warmer climate as a consequence of the Clausius-Clapeyron relation (Held and Soden, 2006), (ii) the lower-tropospheric baroclinicity is expected to decrease in the Northern Hemisphere (NH) because of the enhanced Arctic warming compared to lower latitudes, in particular during the winter season, whereas in the Southern Hemisphere (SH) it is expected to increase (Harvey et al., 2014), and (iii) the upper-tropospheric baroclinicity is expected to increase in both hemispheres in response to a warming of the tropical upper troposphere and a cooling of the polar stratosphere (Collins et al., 2013). These different mechanisms can interact and counteract each other in a complex way and make predictions of future changes in the characteristics of extratropical cyclones challenging. For instance, the increase in the atmospheric moisture content goes along with an increase in the latent heating rate, which may produce more intense cyclones via an enhancement of the low-level cyclonic PV. However, the increased latent heating is also expected to lead to enhanced poleward and upward energy transport, which could in turn decrease the horizontal baroclinicity and the vertical stability of the atmosphere and thereby have a dampening effect on cyclone strength (Catto et al., 2019).

Projections of changes in cyclone properties in a warmer and moister climate depend on the choice of the model, the region, the season and the variables examined (Ulbrich et al., 2009; Catto et al., 2019). In idealised baroclinic lifecycle experiments, Booth et al. (2013) showed that increased atmospheric moisture enhances the cyclone's intensification rate, minimum central pressure, surface winds and precipitation, if the background baroclinicity is not changed. However, in the idealised simulations by Tierney et al. (2018), the cyclone intensities only increased up to a certain temperature threshold beyond which they decreased again. This response to increased temperature and moisture was also found when the baroclinicity was reduced or




increased at the same time. Also Pfahl et al. (2015) and Büeler and Pfahl (2019) documented a non-monotonic response of the median cyclone intensity – measured in terms of minimum central pressure and relative vorticity – to temperature and moisture increases in idealised aquaplanet simulations, with the maximum values at temperatures slightly above the current climate. They explained this non-monotonic behaviour with partly compensating changes in diabatic PV generation, baroclinicity, vertical stability and the tropopause height. In contrast to the moderately intense cyclones, the low-level relative vorticity of the

most extreme ones continued to intensify with increasing temperature and moisture. These cyclones were associated with a considerably stronger enhancement of the latent heating and of the diabatically produced low-level PV anomaly than the moderate cyclones, which overcompensated for the reduction in baroclinicity. In the intense cyclones, both studies also reported an increase in cyclone-related precipitation in warmer climates and, in Büeler and Pfahl (2019), an increase in the near-surface wind speeds that could be related to the increase in the latent heating. Also based on aquaplanet simulations, Sinclair et al.

(2020) found no future changes in the median intensity of the cyclones, as measured in terms of low-level vorticity, and a decrease in their number. The most intense cyclones, on the other hand, increased in number, and they were associated with stronger ascent, low-level PV production and precipitation, and a downstream shift in the diabatic PV anomaly relative to the cyclone centre. Similarly, Kirshbaum et al. (2018) and Rantanen et al. (2019) did not find any increases in cyclone intensity, measured in terms of eddy kinetic energy, with increasing temperature and enhanced diabatic heating. As in Sinclair et al.

(2020), the diabatically produced warm-frontal PV anomaly shifted further downstream in a warmer climate, which led to an unfavourable phasing with the dry upper-level PV anomaly. Thus, in these simulations latent heating rather had a dampening effect on the cyclone evolution. The different studies show that even in idealised settings the response of the cyclones to rising temperatures is not unambiguous and depends on the background state, the exact changes in temperature and moisture and the complex interactions between dry and moist dynamics. It is therefore not surprising that an adequate prediction of future

changes in cyclone dynamics is even more challenging in fully coupled climate models.

Climate models typically project a small reduction in the total number of cyclones in a future climate in both hemispheres (e.g., Bengtsson et al., 2009; Mizuta et al., 2011; Zappa et al., 2013; Grieger et al., 2014; Dolores-Tesillos et al., 2022; Priestley and Catto, 2022). The deepest cyclones, on the other hand, are projected to increase in number in the SH (Grieger et al., 2014; Chang, 2017; Priestley and Catto, 2022). In the NH, there is less consensus in how their number will change, with some studies

indicating a decrease (Zappa et al., 2013; Seiler and Zwiers, 2016; Chang, 2018) and others an increase (Mizuta et al., 2011; Priestley and Catto, 2022). In terms of intensity, a recent analysis of Coupled Intercomparison Project 6 (CMIP6) models projects an increase for the cyclones with peak low-level vorticity in both hemispheres during the winter season (Priestley and Catto, 2022), in agreement with the findings from aquaplanet simulations (Pfahl et al., 2015; Büeler and Pfahl, 2019; Sinclair et al., 2020; Schemm et al., 2022). In extreme winter cyclones, climate models also project an increase in the wind speeds

and in the area of strong winds in the cyclone's warm sector (Dolores-Tesillos et al., 2022; Priestley and Catto, 2022) and an amplification of the diabatically produced low-level PV anomaly (Dolores-Tesillos et al., 2022). Furthermore, it is widely agreed that cyclone-related precipitation will increase with global warming both in moderate and in intense cyclones as a result of the increased atmospheric water vapour (Bengtsson et al., 2009; Hawcroft et al., 2018; Raible et al., 2018). Finally, the storm track is expected to shift poleward in the North Pacific during winter and in the SH during both winter and summer (e.g., Yin,



2005; Bengtsson et al., 2009; Priestley and Catto, 2022), which can largely be explained by the increased latent heating and diabatic PV production (Tamarin-Brodsky and Kaspi, 2017).

The idealised simulations as well as the climate model studies demonstrate that the role of enhanced diabatic heating for cyclone development in a warming climate is not yet fully understood. As the strongest diabatic heating in extratropical cyclones occurs in WCBs, the question arises how WCB-related diabatic PV production changes with global warming. In an accompanying study by Joos et al. (2022), a climatology of WCB trajectories has been calculated for the first time in climate model simulations with the Community Earth System Model Large Ensemble (CESM-LE; Kay et al., 2015), and their geographical distribution, seasonal frequency and characteristics have been investigated in the present-day and end-of-century climate. It was shown that the present-day simulations are able to realistically capture the geographical distribution and frequency of occurrence of the WCBs in many regions. In the future simulations, overall the WCB frequency maxima are located in similar regions as in the present climate, but there are also some geographical shifts, and the total number of WCB trajectories increases. In regional WCB hotspots like the North and South Atlantic, North Pacific and Mediterranean, there is an increase in the WCB inflow moisture and in the precipitation and the diabatic heating rate along the ascending trajectories, and the maximum in the diabatic heating rate shifts upward.

The projected overall increase in the number of WCB trajectories and in the diabatic heating rate along the trajectories in a warmer climate potentially influences the development of the associated cyclones. In this study, we use the same WCB climatology as in Joos et al. (2022) and combine it with a cyclone climatology to evaluate how WCBs and their associated diabatic PV production affect cyclone intensification in present-day and future CESM-LE simulations during NH and SH winter. In addition to a climatological analysis of all cyclones, we investigate potential future changes in the characteristics of a subset of cyclones with particularly strong WCBs and explosive deepening, referred to as C1 cyclones (see Binder et al., 2016). Specifically, the following research questions are addressed:

1. Is CESM-LE able to capture the properties and structure of extratropical cyclones and their associated WCBs when compared to reanalysis data?

2. How will the WCB strength and the cyclone deepening rate change in a future climate according to the representative concentration pathway 8.5 (RCP8.5) emission scenario, and how will the number of explosive cyclones in general and those with strong WCBs change in the two hemispheres?

3. How will climate change affect the importance of WCB-related diabatic PV production for cyclone intensification, and how will it affect the properties, structure and intensity of C1 cyclones?

Section 2 describes the climate model and reanalyses used in the study, along with the methods to identify WCBs and cyclones and to combine the two data sets. In Section 3, CESM-LE is evaluated by comparing the present-day simulations with reanalysis data. Section 4 presents results of future changes in WCB strength, cyclone deepening rate and cyclone number, and of future changes in the characteristics of C1 cyclones. A summary and conclusions are provided in Section 5.





## 2 Data and Methods

### 2.1 Climate simulations and reanalysis data

The study is based on output from an initial condition ensemble of the Community Earth System Model (CESM), version 1
(Hurrell et al., 2013). The ensemble data was created by rerunning the CESM Large Ensemble (CESM-LE; Kay et al., 2015)
with restart files from the original simulations (see Röthlisberger et al., 2020, for a detailed description). The re-simulations
were produced to obtain high-resolution three-dimensional model level output, such as wind fields, which are required for
the computation of WCB trajectories. The data is available every 6 h at a horizontal resolution of $1.25°$ longitude by $\sim 0.9°$
latitude and on 30 vertical levels. Two time periods are simulated, 1990-1999 for the present-day climate (hereafter referred to
as CESM-HIST), which is based on historical forcing (Lamarque et al., 2010), and 2091-2100 for the future climate (hereafter
referred to as CESM-RCP85), based on the RCP8.5 emission scenario (Lamarque et al., 2011; Meinshausen et al., 2011). Both
time periods are simulated with five ensemble members, which only differ by small perturbations in the initial atmospheric
temperature field. In total, this yields 50 simulated years for each time period. The analysis is confined to the winter season in
both hemispheres, i.e., December-February for the NH and June-August for the SH.

To validate the ability of CESM-LE to represent extratropical cyclones and their associated WCBs, the present-day simula-
tions are compared to the corresponding fields in ERA-Interim reanalyses of the European Centre for Medium-Range Weather
Forecasts (Dee et al., 2011) for the period 1979-2014. The ERA-Interim data set is the same as in Binder et al. (2016). The
variables are available every 6 h on 60 levels in the vertical, and they have been interpolated from a spectral resolution of T255
to a $1°$ by $1°$ horizontal grid. As for the CESM simulations, we restrict the analysis to the winter season in both hemispheres.

### 2.2 Cyclone and WCB identification

Extratropical cyclones are identified based on the algorithm of Wernli and Schwierz (2006) that has been refined in Sprenger
et al. (2017). Cyclones are defined as two-dimensional features delimited by the outermost closed isobar that encompasses a
local sea level pressure (SLP) minimum or several minima. In addition, a tracking algorithm determines for each SLP minimum
the most probable continuation among the minima identified 6 h later and thereby follows the cyclones from genesis to maturity
to lysis. The analysis is restricted to cyclones with a lifetime of at least 48 h, and to exclude tracks with unrealistic lifecycles,
the pressure difference between the outermost closed SLP contour and the SLP minimum at the beginning of the track needs
to be less than 5 hPa (see Binder et al., 2016, for details). Furthermore, we only consider cyclones located poleward of $25°N$
and S during at least a 24 h interval, to exclude tropical cyclones.

    The identification of WCB trajectories is the same in ERA-Interim and the CESM simulations and is described in detail
in Madonna et al. (2014) and Joos et al. (2022), respectively. Using the Lagrangian analysis tool (LAGRANTO; Wernli and
Davies, 1997; Sprenger and Wernli, 2015), 48 h forward trajectories are started every 6 h from a horizontally equidistant grid
with $\Delta x = 80$ km resolution and from vertically equidistant ($\Delta p = 20$ hPa) levels in the lower troposphere (1050-790 hPa).
To be selected as WCB air parcels, the trajectories must undergo an ascent of at least 600 hPa within 48 h. In addition, their
horizontal position must coincide with the surface field of an extratropical cyclone at least once during the 48 h.





A technical problem occurs at the end of the years: In CESM, we treat each simulated year separately, such that the trajectories are not able to cross the year end. This implies that the last four days of each year are not included in the WCB climatology. However, we expect that this only has a negligible impact on the results.

### 2.3   Link between cyclone deepening and WCB strength

To study the link between cyclone deepening and WCB strength (as defined below), the WCB trajectories are assigned to
cyclones. In ERA-Interim, each WCB trajectory is assigned to the first cyclone with which it overlaps during the 48 h ascent, as in Binder et al. (2016). In CESM-LE, WCB trajectories located close to each other are first clustered with a similar clustering algorithm as the one described in Catto et al. (2015b). All trajectories in the cluster are attributed to the same cyclone, namely the cyclone with which the highest number of WCB trajectories overlaps when they are located between 850 and 500 hPa. Both in ERA-Interim and in CESM-LE this yields for every cyclone during the entire lifecycle the number of associated WCB
trajectories and their properties. The slightly different procedures to link WCB trajectories with cyclones is expected to only negligibly influence the results. Indeed, the number of WCB trajectories per cyclone is very similar in ERA-Interim and the present-day simulations of CESM-LE, as will be discussed in Section 3.

Cyclone deepening is measured in Bergeron units (Sanders and Gyakum, 1980). Over each 24 h interval during a cyclone lifecycle, the latitude-adjusted SLP deepening within 24 h is calculated as follows:

$$\Delta SLP_B \, [\text{Bergeron}] \; = \frac{\Delta SLP \, [\text{hPa}]}{24 \, \text{h}} \cdot \frac{\sin(60°)}{\sin(\phi)},$$

where $\Delta SLP$ is the change in the minimum SLP and $\phi$ represents the cyclone's average latitude during the 24 h time period. For each cyclone, the deepening rate $\Delta SLP_{B,max}$ is then determined as the largest value in $\Delta SLP_B$ over all 24 h intervals during the cyclone lifecycle. Sanders and Gyakum (1980) defined explosively intensifying cyclones (so-called "bombs") as cyclones with a deepening rate of more than 1 Bergeron, which corresponds to a SLP drop of more than 24 hPa in 24 h at 60°
latitude.

The cyclone's WCB strength is quantified by the number of WCB trajectories assigned to that cyclone and located in the lower troposphere (pressure > 500 hPa) at any time during the 24 h of strongest deepening. The trajectories do not necessarily need to be located within the cyclone area during these 24 h, as they or – for CESM-LE – part of their cluster can overlap with the cyclone area at any time during their two-day ascent. Since the WCB trajectories were started from an equidistant grid (see
Section 2.2), each trajectory is associated with the same air mass, i.e., $\Delta m \approx \frac{1}{g}(\Delta x)^2 \Delta p \approx 1.3 \times 10^{12}$ kg, with $g = 9.81 \, \text{m s}^{-2}$. Thus, multiplication of the number of WCB trajectories with $\Delta m$ yields the WCB air mass.

### 3   Climate model evaluation

To have confidence in CESM-LE's projections of extratropical cyclones and their associated WCBs in a warming climate, the model needs to be able to correctly represent them in the current climate. Joos et al. (2022) compared the WCB climatology

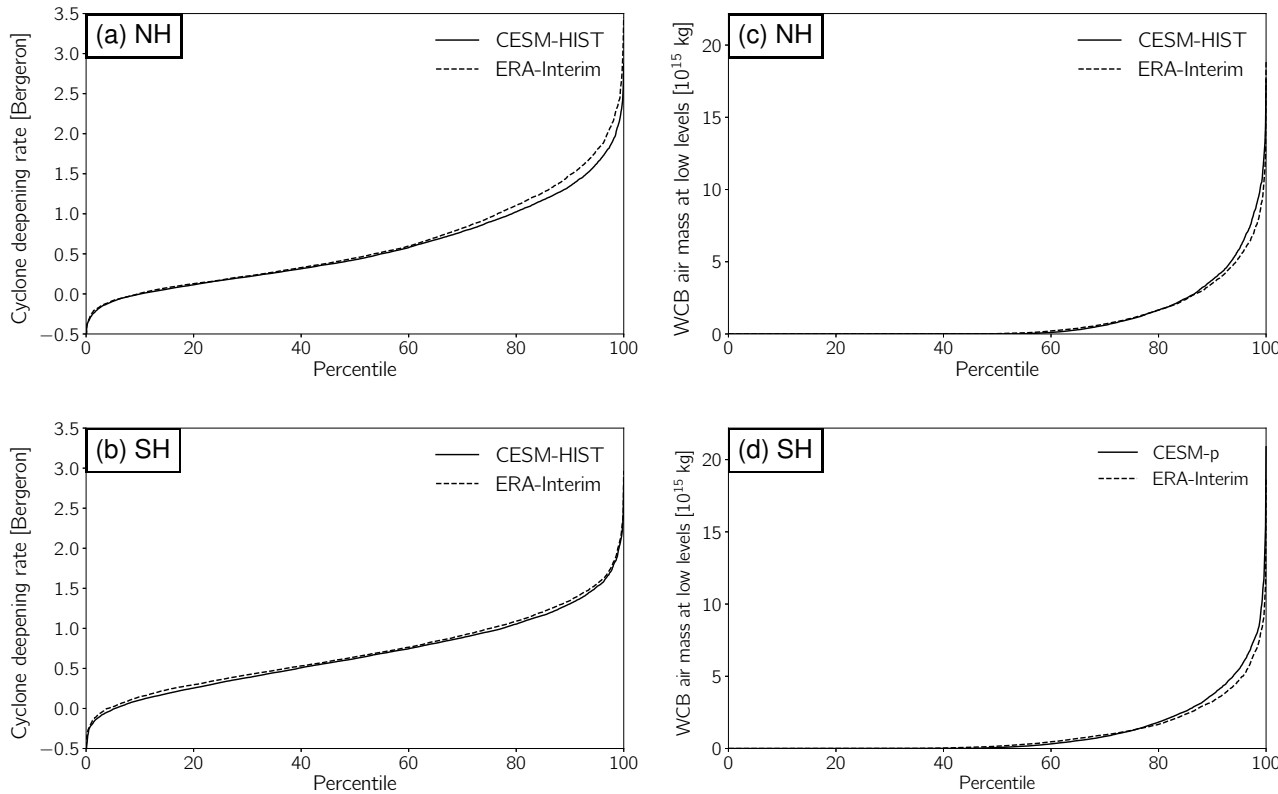

**Figure 1.** Percentile curves of the (a, b) cyclone deepening rates $\Delta SLP_{B,max}$ (Bergeron) and (c, d) WCB strength associated with the cyclones (measured by the WCB air mass at $p > 500$ hPa during the cyclone's 24 h of strongest intensification, in $10^{15}$kg) for all cyclones in (a, c) NH winter and (b, d) SH winter. The curves are shown in solid lines for CESM-HIST and dashed lines for ERA-Interim.

in CESM-HIST with ERA-Interim. They showed that in most regions CESM-HIST is able to realistically simulate WCBs in terms of their geographical distribution, frequency of occurrence and ascent behaviour. For the North Atlantic, Raible et al. (2018) and Dolores-Tesillos et al. (2022) showed that the model captures cyclone frequencies and lifetimes relatively well. In this Section, we evaluate some further characteristics of WCBs, cyclones and the link between them in CESM-HIST and ERA-Interim. In CESM-HIST, the results are based on the years 1990-1999. In ERA-Interim, they are based on the years 1979-
2014 as in Binder et al. (2016), but the results are very similar when only considering the years 1990-1999 (not shown). Note that we do not expect an exact agreement between ERA-Interim and the climate simulations. The two data sets are based on two different models, with, for instance, different spatial resolution and different parameterization schemes for sub-grid-scale processes. In addition, ERA-Interim only represents one possible realisation of the present-day climate, whereas CESM-HIST represents an ensemble with equally likely realisations of the same decade.

CESM-HIST slightly underestimates the number of extratropical cyclones per winter in the NH (132 cyclones per winter in CESM-HIST vs. 141 cyclones in ERA-Interim), but performs very well in the SH (136 cyclones in both data sets; Table 1).





Figure 1a shows, for NH winter, percentile curves of the intensification rates of all cyclones ($\Delta SLP_{B,max}$), separately for CESM-HIST and ERA-Interim. Hereby, the curve indicates the percentage of cyclones (with respect to the total number as indicated in Table 1) with a specific intensification rate or lower. For instance, 90% of the cyclones have a deepening rate

$\leq 1.35$ Bergeron in CESM-HIST and $\leq 1.5$ Bergeron in ERA-Interim, while the remaining 10%, i.e., the 10% most strongly deepening cyclones, have Bergeron values above this threshold. The agreement between CESM-HIST and ERA-Interim is very good up to approximately the 70th percentile; for higher percentiles CESM-HIST underestimates the intensification rates by about 0.1 to 0.3 Bergeron. In the SH, the agreement is good for all percentiles, with discrepancies around 0.05 Bergeron or less (Fig. 1b).

In Figs. 1c and d, percentile curves of the WCB strength associated with the cyclones are displayed for NH and SH winter, respectively. In both hemispheres, about half of the cyclones do not have a WCB in ERA-Interim and CESM-HIST. For cyclones with WCB trajectories, CESM-HIST slightly overestimates the WCB strength for the highest percentiles, but overall it performs remarkably well.

Figure 2a shows for all NH winter cyclones in ERA-Interim a two-dimensional phase-space of the link between WCB

strength and cyclone intensification rate, equivalent to Fig. 1 in Binder et al. (2016). In Fig. 2c, the same is shown for CESM-HIST. Consistent with Fig. 1, the deepening rates of the most explosive cyclones are higher in ERA-Interim. Otherwise, the cyclones are distributed fairly similarly in the phase spaces in the reanalyses and the climate model. In both plots, the red and light green colours reveal that most of the cyclones with no or few WCB trajectories also experience a weak deepening. Dark green and blue colours indicate that cyclones with a more intense WCB typically deepen more strongly. The Spearman rank

correlation coefficient has the same value of 0.68 in ERA-Interim and CESM-HIST, which corresponds to a moderate to strong positive correlation between WCB strength and cyclone intensification rate. In the SH, the Spearman correlation coefficients are lower (0.46 in ERA-Interim and 0.51 in CESM-HIST), but the distribution of the cyclones in the phase-space is also well captured by the climate model (Fig. 2b,d).

To assess whether CESM-HIST is able to adequately represent the structure and properties of extratropical cyclones and the

associated WCBs, we investigate the average fields of a subset of cyclones with a compositing method. We restrict our analysis to cyclones with particularly strong deepening and intense WCBs, referred to as C1 cyclones. The category borders of the C1 cyclones are displayed in Fig. 2. They are identical to those defined in Binder et al. (2016), i.e., explosively deepening cyclones ($> 1$ Bergeron) with a WCB intensity of at least $2.78 \times 10^{15}$ kg (corresponding to 2130 WCB trajectories). In ERA-Interim, this yields 500 C1 cyclones in the NH, which corresponds to 9.9% of the total cyclone number, and 330 C1 cyclones in the

**Table 1.** Number of wintertime extratropical cyclones in ERA-Interim, CESM-HIST and CESM-RCP85.

| Number of cyclones (total / per winter) | NH | SH |
|---|---|---|
| ERA-Interim (36 winter) | 5069 / 141 | 4890 / 136 |
| CESM-HIST (50 winter) | 6578 / 132 | 6807 / 136 |
| CESM-RCP85 (50 winter) | 6424 / 128 | 6294 / 126 |



**Figure 2.** Two-dimensional histograms showing for all (a, c, e) NH and (b, d, f) SH winter cyclones in (a, b) ERA-Interim, (c, d) CESM-HIST, and (e, f) CESM-RCP85 the cyclone intensification rate $\Delta SLP_{B,max}$ (bin width = 0.1 Bergeron) and the WCB strength (bin width ≈ $0.33 \times 10^{15}$ kg, corresponding to 250 WCB trajectories). The colours specify the number of cyclones per bin. The black lines mark the borders of categories C1, C2 and C3.





SH, corresponding to 6.7%. In CESM-HIST, the larger number of simulated years goes along with more C1 cyclones (616 in the NH and 509 in the SH), but the fraction from the total cyclone number is with 9.4% and 7.5%, respectively, very similar to the one in ERA-Interim. The composites are created in the middle of the 24 h interval of strongest cyclone intensification and centred at the location of the SLP minimum.[1] Note that here we are mainly interested in the differences between CESM-HIST and ERA-Interim, a detailed description of the structure and evolution of the C1 composite cyclone in ERA-Interim for NH

winter can be found in Binder et al. (2016). Here we only show C1 composites for NH winter, but the findings are very similar for SH winter.

Figure 3a,b shows SLP, PV at 250 hPa and the WCB frequency of the composite cyclones for ERA-Interim and CESM-HIST, respectively. For ERA-Interim, the figure is almost identical to Fig. 7c in Binder et al. (2016). The minimum SLP has a value of about 988 hPa in ERA-Interim and 987 hPa in CESM-HIST. Both in the reanalysis and the climate model, a cyclonically

breaking upper-level disturbance is located to the west of the surface cyclone, with a PV gradient of very similar magnitude. Low-level (upper-level) WCB frequencies indicate the fraction of C1 cyclones that has at least one WCB trajectory located at $p > 500$ hPa ($p < 500$ hPa) at a specific position. The spatial distribution of the frequencies agrees well between reanalysis and climate model. In both cases, high low-level WCB frequencies occur in the warm sector and above the warm front of the composite cyclone, with maximum values of more than 80% close to the cyclone centre. The highest upper-level WCB

frequencies amount to about 60% and are located slightly to the north of the peak low-level values. Additionally shown is the frequency of low-level WCB air parcels located between 900 and 700 hPa that reach high PV values above 1 pvu (HLPV WCB frequencies, green contours), which highlights the regions with most pronounced WCB-related diabatic PV production. The HLPV trajectories are co-located with the cyclone centre and their maximum frequencies amount to approximately 65% in ERA-Interim and 55% in CESM-HIST.

The potential temperature structure and the horizontal temperature gradient at 850 hPa are very similar in ERA-Interim and CESM-HIST (Fig. 3c,d). At upper levels, in both data sets a jet streak is located southwest of the cyclone centre. Also the precipitation pattern agrees well between ERA-Interim and CESM-HIST, with non-zero values in the entire warm sector (Fig. 3c,d). Maximum values near the cyclone centre amount to 17 mm h$^{-1}$ in ERA-Interim and 16 mm h$^{-1}$ in CESM-HIST. In both cases, they coincide with the region of strongest WCB-related PV production (green contours in Fig. 3a,b). The higher

maximum precipitation values and HLPV WCB frequencies in ERA-Interim are reflected in a deeper and more pronounced diabatically produced low-level PV anomaly, as evident in the cross section across the cyclone centre (Fig. 3e,f). The maximum PV values in the cyclone centre reach about 1.5 pvu in ERA-Interim and 1 pvu in CESM-HIST. This relatively strong underestimation in the climate model, which also occurs in the SH (not shown), could be associated with the lower spatial resolution, in particular in the vertical (30 vertical levels in CESM-LE compared to 60 in ERA-Interim), as well as differences in the rep-

resentation of moist processes in the two parameterisation schemes. At upper levels, CESM-HIST represents the PV structure well, with a deep trough upstream of the cyclone centre and a ridge downstream (Fig. 3e,f). During the subsequent hours, in both data sets continuous WCB-related PV formation and a cyclonic wrapping-up of the disturbance at the tropopause con-

---

[1]To ensure that all grid points are associated with a similar area, previous to the compositing a coordinate transformation is performed, whereby the cyclone centres are shifted to the equator and 0° longitude and then the mean fields are calculated at these positions.



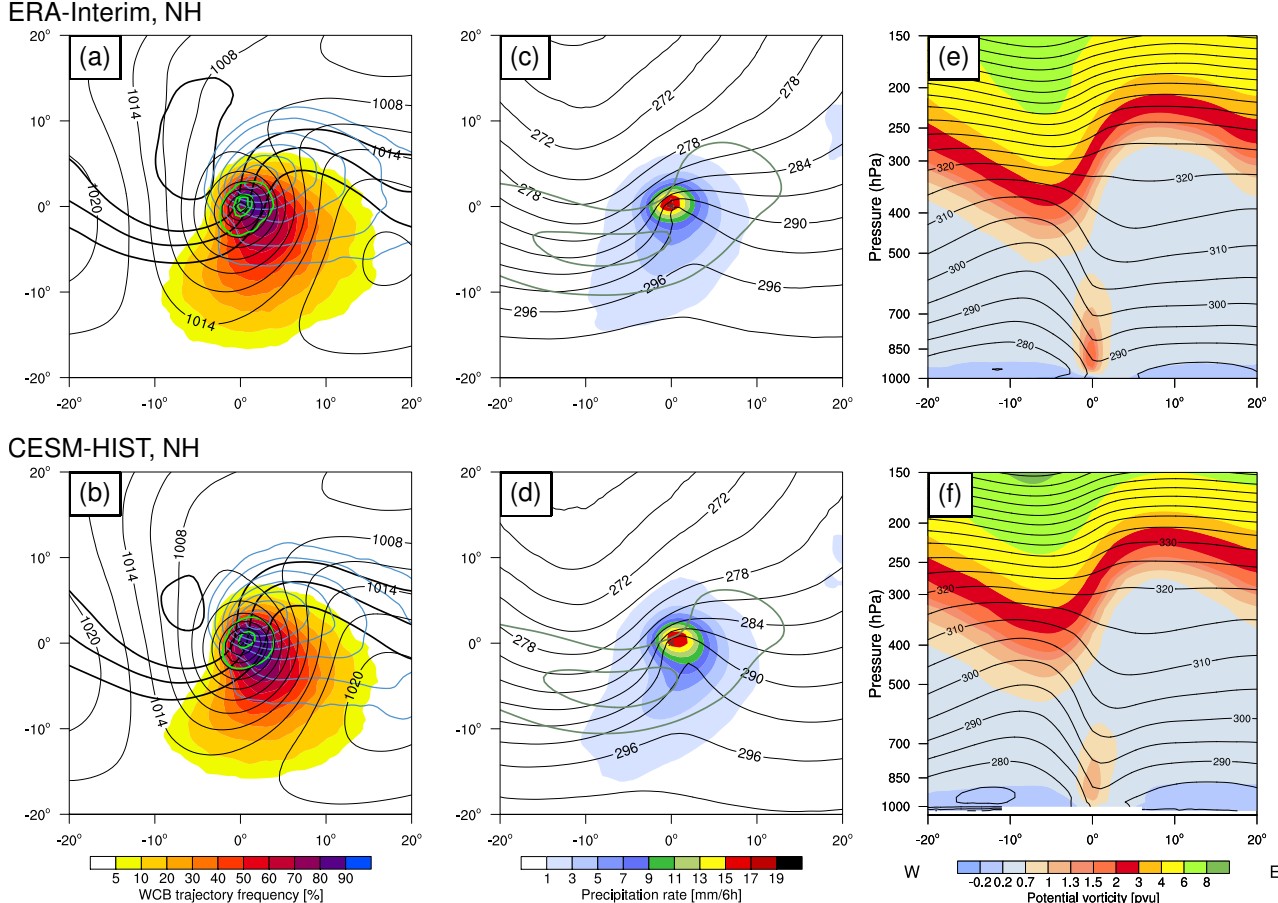

**Figure 3.** Composites of C1 cyclones in NH winter in (a, c, e) ERA-Interim and (b, d, f) CESM-HIST in the middle of the 24 h period of strongest deepening. (a, b) Frequency of WCB air parcels with $p > 500$ hPa (%; shading), $p < 500$ hPa (blue contours at 5%, 10%, 20%, 30%, ... , 80%) and WCB air parcels with PV $> 1$ pvu and $900 > p > 700$ hPa (green contours at 10%, 50% and 60%), SLP (thin black contours every 3 hPa) and PV at 250 hPa (thick black contours at 2, 3, 4 and 6 pvu). (c, d) 6-h accumulated precipitation (mm; shading), potential temperature at 850 hPa (black contours every 3 K) and wind speed at 250 hPa (green contours at 50 and 60 m s$^{-1}$). (e, f) West-east oriented cross section across the centre of the composite cyclone, showing PV (pvu; shading) and potential temperature (black contours every 5 K).

tribute to the further intensification of the composite cyclone, until 12 h later, at the end of the strongest intensification phase, the positive low- and upper-level PV anomalies align vertically and form a troposphere-spanning PV tower (Hoskins, 1990;
Rossa et al., 2000; Badger and Hoskins, 2001), which is associated with an intense cyclonic wind field (for ERA-Interim, see Fig. 7f in Binder et al., 2016). Still, the diabatically produced PV anomaly within the PV tower is about 0.5 pvu lower in CESM-HIST than in ERA-Interim (not shown).



In summary, this comparison between ERA-Interim and CESM-HIST showed that CESM-HIST is able to capture the properties and structure of extratropical cyclones and their associated WCBs remarkably well. There are some differences between
climate model and reanalysis, like, for instance, a weaker deepening rate of the most explosive cyclones in CESM-HIST in the NH and a weaker intensity of the diabatically produced positive PV anomaly in the centre of the C1 cyclones. The overall very good representation of extratropical cyclones and WCBs in the climate model gives confidence in the model's ability to predict potential changes in a warming climate. In the following, we compare CESM-LE simulations of the present-day and the future climate, to assess how climate change affects the cyclones' WCB strength and the importance of WCB-related diabatic PV
production for cyclone intensification.

## 4 WCBs and cyclones in a warming climate

### 4.1 Future changes in cyclone number, cyclone intensification rate and WCB strength

The number of cyclones per winter is projected to decrease in the future climate by 3% in NH winter (from 132 to 128 cyclones) and 7% in SH winter (from 136 to 126 cyclones; Table 1). This is consistent with the findings from many previous studies (e.g.,
Bengtsson et al., 2009; Grieger et al., 2014; Priestley and Catto, 2022). To evaluate potential changes in cyclone deepening rates and WCB strengths in the future climate, we look at the differences in the percentile curves between the future and the present-day climate simulations (CESM-RCP85 minus CESM-HIST; Fig. 4). Positive differences for specific percentiles indicate an increase and negative differences a decrease in the deepening rates (Fig. 4a,b) and WCB strengths (Fig. 4c,d) in the future climate. To detect statistically significant differences, a confidence interval has been constructed by performing
a bootstrap resampling. Hereby, our null hypothesis was that the cyclone intensification rates and the WCB strengths per cyclone, respectively, do not differ between the present-day and future climate simulations, i.e., that they belong to a common distribution. From this common distribution, 100'000 resamples have been created by randomly attributing half of the values to CESM-HIST and half of the values to CESM-RCP85 and then calculating the difference in the percentile curves. The 2.5th and the 97.5 percentiles of the ranked differences of the resampling distribution correspond to the lower and upper limits of
the 95% confidence interval. If the actual difference lies outside this interval, it is statistically significant with a probability of 95%.

In NH winter, the cyclone intensification rate is projected to increase slightly in the future climate for the weakest as well as the most rapidly intensifying cyclones and to decrease for the 50th-90th percentile, but the changes are smaller than 0.05 Bergeron and not statistically significant for almost all percentiles (Fig. 4a). In SH winter, there is a statistically significant
increase in the intensification rates by about 0.03-0.05 Bergeron for medium-strong cyclones (70th-90th percentile) and by about 0.05-0.1 Bergeron for strongly intensifying cyclones above the 90th percentile (Fig. 4b). Intensification rates typically range from about $-0.5$ to 3 Bergeron (Fig. 2), such that an increase by 0.05-0.1 Bergeron for the most strongly intensifying cyclones corresponds to a future change by about 1.5%-3%. The small changes in the intensification rates in the NH and the increase for the most extreme cyclones in the SH are in line with previous studies that have assessed future changes in cyclone
strength based on different measures (e.g., Catto et al., 2019; Priestley and Catto, 2022).

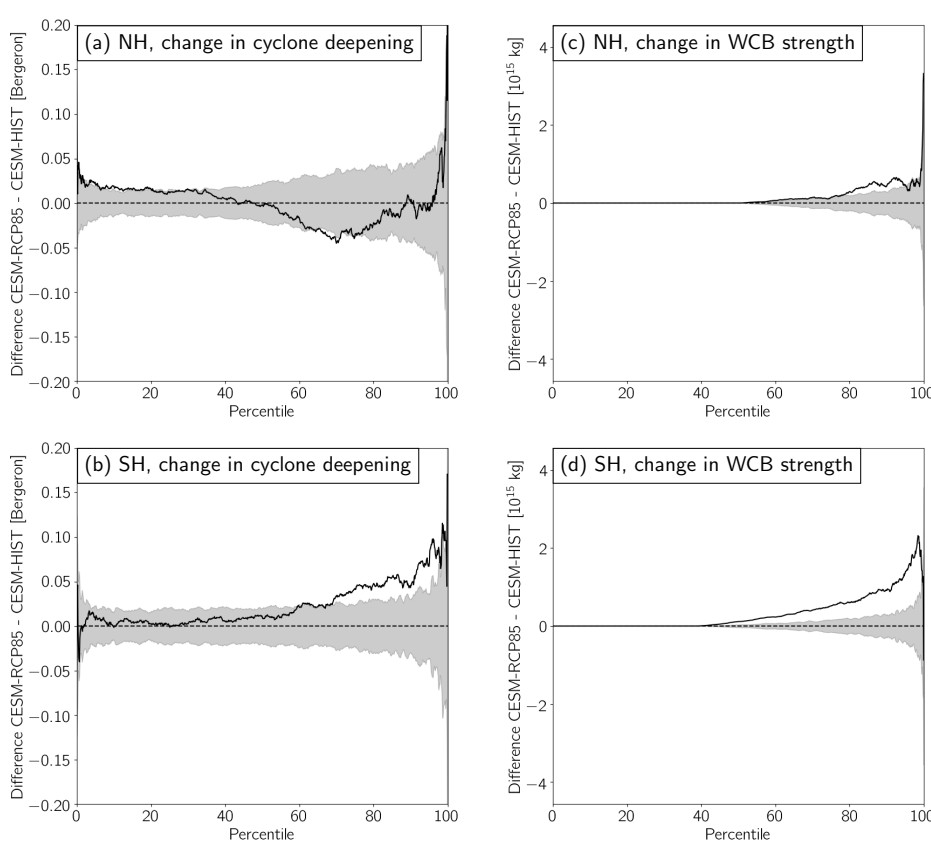

**Figure 4.** The black line shows the difference in the percentile curves between the future and the present-day climate simulations (CESM-RCP85 minus CESM-HIST) of (a, b) cyclone deepening rates $\Delta SLP_{B,max}$ (Bergeron) and (c, d) WCB strength associated with the cyclones ($10^{15}$ kg) in (a, c) NH winter and (b, d) SH winter. The grey shading corresponds to the 95% confidence interval of the differences under the null hypothesis that the values in CESM-HIST and CESM-RCP85 belong to a common distribution. It has been obtained by calculating 100'000 resamples of the difference in the percentile distribution from two randomly drawn equal-sized samples (see text for details).



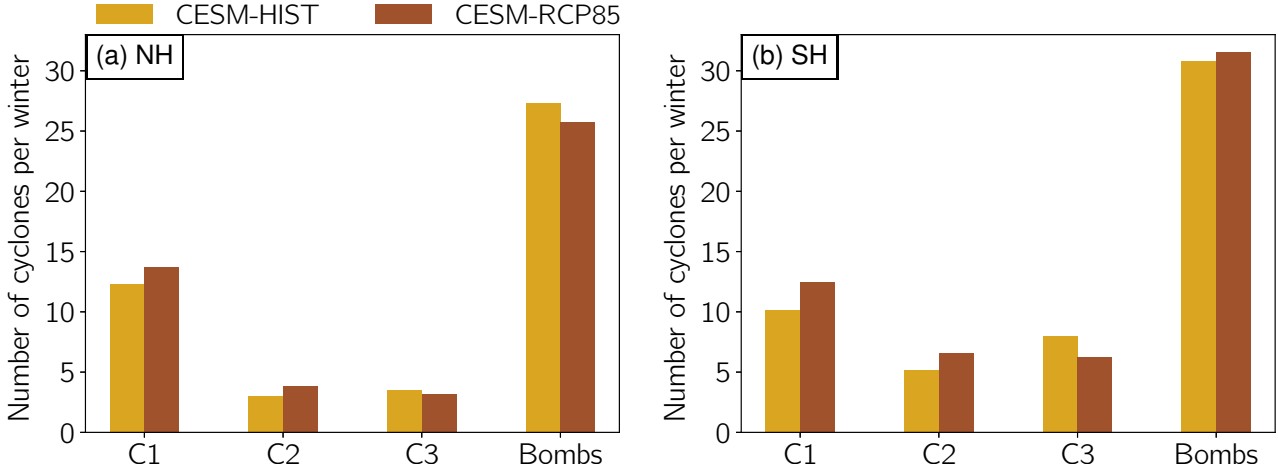

**Figure 5.** Number of C1, C2, C3 and bomb cyclones per winter in CESM-HIST and CESM-RCP85 in the (a) NH and (b) SH.

In both hemispheres, in cyclones that are associated with WCBs, the WCBs become stronger in the future climate (Fig. 4c,d), consistent with the findings from Joos et al. (2022) (their Table 1). In the SH, the changes are statistically significant for almost all percentiles (Fig. 4d). The largest increases amount to about $2 \times 10^{15}$ kg, which corresponds to 1500 WCB trajectories per cyclone or 11%. In the NH, the changes are much smaller and only statistically significant for the 80th-95th percentile (Fig. 4c).

There, the increases amount to about $0.4 - 0.6 \times 10^{15}$ kg, i.e., 300-460 WCB trajectories per cyclone or 2-3%

The significant increase in the cyclone intensification rate in the SH but not in the NH could in part be due to the much stronger increase in the WCB strength and accordingly the WCB-related latent heating in the SH. In addition, the differences in the intensification rates between the two hemispheres are consistent with the opposite changes in low-level baroclinicity expected with global warming (e.g., Harvey et al., 2014; Catto et al., 2019, see also introduction): In the SH, low-level baro-

clinicity is expected to increase, which favourably interacts with the increased WCB strength and leads to stronger cyclones, whereas in the NH it is expected to decrease, which counteracts the effects of the increased WCB strength such that the cyclone intensity does not change considerably or even decreases.

The statistical correlation between the cyclone deepening rate and the WCB strength is shown in the two-dimensional histograms in Fig. 2c-f. In both hemispheres, cyclone deepening correlates positively with the intensity of the associated

WCB in the present-day and future climate simulations, indicating that WCBs continue to play an important role for cyclone intensification in a warming climate. In the NH, the Spearman correlation coefficient has a value of 0.68 in CESM-HIST (Fig. 2c) and 0.66 in CESM-RCP85 (Fig. 2e). In the SH, it increases from 0.51 in CESM-HIST (Fig. 2d) to 0.55 in CESM-RCP85 (Fig. 2f).

Despite the overall decrease in the number of cyclones per winter in both hemispheres (Table 1), the number of C1 cyclones,

i.e., explosively deepening cyclones with strong WCBs, is projected to increase (Fig. 5). In the NH, it increases by 11% from 12.3 to 13.7 C1 cyclones per winter (Fig. 5a) and in the SH by 23% from 10.2 to 12.5 cyclones (Fig. 5b). In both hemispheres, there is also a ~25% increase in the percentage of so-called C2 cyclones, which are defined as in Binder et al. (2016) to have





a similar WCB strength as C1 (at least $2.78 \times 10^{15}$ kg) but a weak deepening rate of less than 0.8 Bergeron (see Fig. 2 for the category boundaries). At the same time, the percentage of C3 cyclones, i.e., explosively intensifying cyclones with weak

WCBs of less than $0.33 \times 10^{15}$ kg, decreases by 11% in the NH and 21% in the SH. The total number of explosive cyclones irrespective of the associated WCB strength (so-called bombs) decreases by 6% in the NH and increases by 2% in the SH. The changes in the phase-space diagram in Fig. 2, i.e., the increase in the number of C1 and C2 cyclones and the decrease in C3, are in line with the overall increase in the cyclone-related WCB strength observed in Fig. 4c,d and indicate that cyclones become more diabatic in a future climate.

To sum up, in the SH CESM-LE projects an increase in the WCB strength, the cyclone intensification rate and the correlation between the two in the future climate winter. An increase is also projected in the total number of explosive cyclones and those with strong WCBs (C1), and in the number of weak cyclones with strong WCBs (C2), whereas the number of "dry bombs" with weak WCBs (C3) decreases. In the NH, the WCB strength is also projected to increase, but to a smaller extent than in the SH, and there are no significant changes in the cyclone intensification rates. While the number of C1 and C2 cyclones

increases, there is a decrease in the total number of explosive cyclones and in the number of C3 cyclones. In the following, we investigate whether the C1 cyclones themselves will change with regard to spatial distribution, structure and temporal evolution, in addition to their increase in number.

### 4.2 Future changes in the characteristics of C1 cyclones

At the beginning of their strongest intensification, both in the present-day and future climate simulations most C1 cyclones

are located in the western and central parts of the NH and SH oceans, where WCBs are particularly frequent (not shown). However, in the future simulations, the intensification starts about 1.5° farther poleward in the NH and 2° farther poleward in the SH (Table 2). In both hemispheres, the poleward shift is relatively small in the Atlantic and larger in the Pacific basin. The shift is in agreement with previous studies (e.g., Yin, 2005; Bengtsson et al., 2009; Priestley and Catto, 2022).

The time evolution of minimum SLP and the WCB air mass in the lower troposphere shows for both simulations the explosive

deepening of the C1 cyclones and, concomitantly, a rapid increase in the WCB strength, which peaks during the strongest deepening phase (Fig. 6a,b). Consistent with Fig. 4c,d, in both hemispheres but especially in the SH the peak WCB intensity is higher in the future than in the present-day simulations. At the same time, in the future simulations the deepening rate of the C1 cyclones increases slightly and the minimum SLP at the time of the cyclone's strongest intensity is 2 hPa deeper in the

**Table 2.** Mean latitude of C1 cyclones at the beginning of their strongest deepening

| Mean latitude | NH | SH | NATL | NPAC | SATL | SPAC |
|---|---|---|---|---|---|---|
| CESM-HIST | 37.0°N | 36.9°S | 40.2°N | 35.4°N | 39.8°S | 32.8°S |
| CESM-RCP85 | 38.5°N | 39.0°S | 40.9°N | 37.2°N | 40.3°S | 33.8°S |



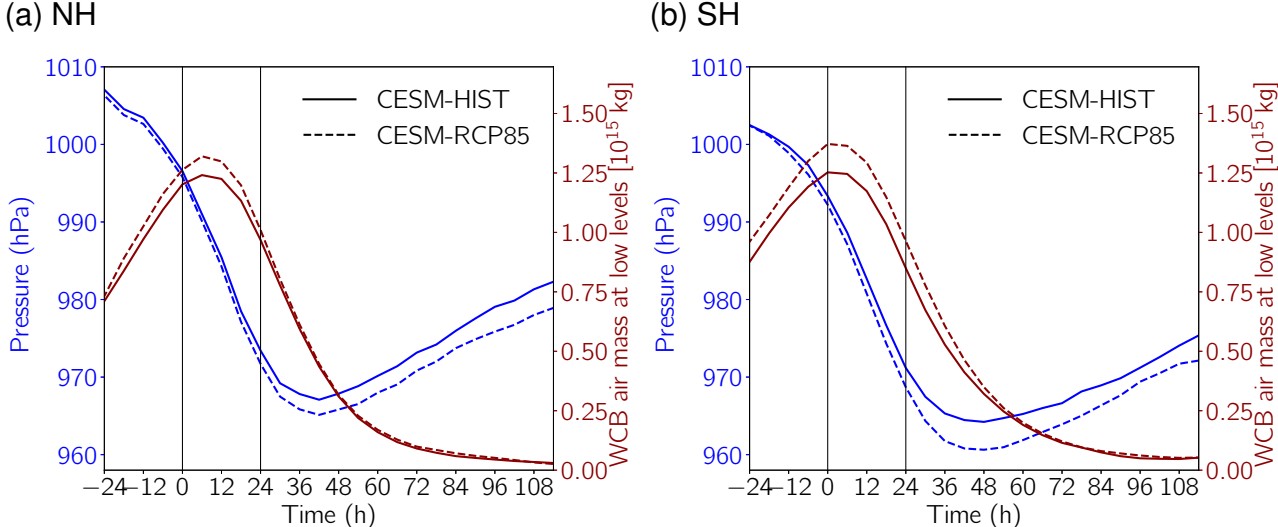

**Figure 6.** Mean time evolution of minimum SLP (hPa; blue) and the WCB air mass at $p > 500$ hPa (red; $10^{15}$ kg) along C1 cyclones in CESM-HIST (solid) and CESM-RCP85 (dashed) in the (a) NH and (b) SH.

NH and 4 hPa deeper in the SH. Thus, in addition to their increase in number (Fig. 5), C1 cyclones also become slightly more
intense in a warming climate.

Figures 7 and 8 show the composite structure of the C1 cyclones in the NH and SH, respectively, in the middle of the 24 h period of strongest intensification, in the present-day and future climate simulations. The fields in Fig 7c,e,g for CESM-HIST are the same as those in the previously discussed Fig 3. Comparison of the cyclones in the two hemispheres in the present-day simulations shows that they have a quite similar structure during the deepening phase. Both in the NH and in the SH, the
composite cyclone is situated ahead of an intense upper-level trough in the left exit region of a jet maximum (Figs. 7c,e,g and 8c,e,g). However, in the NH the PV gradient along the trough is stronger and the jet more intense. Also the equator-to-pole potential temperature gradient at 850 hPa is stronger than in the SH (cf. Figs. 7e and 8e). On the other hand, C1 cyclones in the SH are associated with higher low-level humidity, a larger WCB frequency, higher diabatically produced low-level PV and deeper SLP (cf. Figs. 7a,c,g and 8a,c,g).

The composite cyclones' surroundings become significantly warmer and moister in the future climate. In the cyclone centre, potential temperature, equivalent potential temperature and specific humidity at 850 hPa increase by about 3 K, 6 K and 1.5-2 g kg$^{-1}$ in both hemispheres. (Figs. 7a,b,e,f and 8a,b,e,f). The relative humidity at 850 hPa, on the other hand, does not change considerably near the cyclone centre, as it is already close to saturation in CESM-HIST (Figs. 7a,b and 8a,b). The equator-to-pole potential temperature gradient at 850 hPa in the environment of the C1 cyclones decreases slightly in the NH (Fig. 7e,f)
and increases slightly in the SH (Fig. 8e,f), consistent with the overall changes in low-level baroclinicity projected for the two hemispheres.

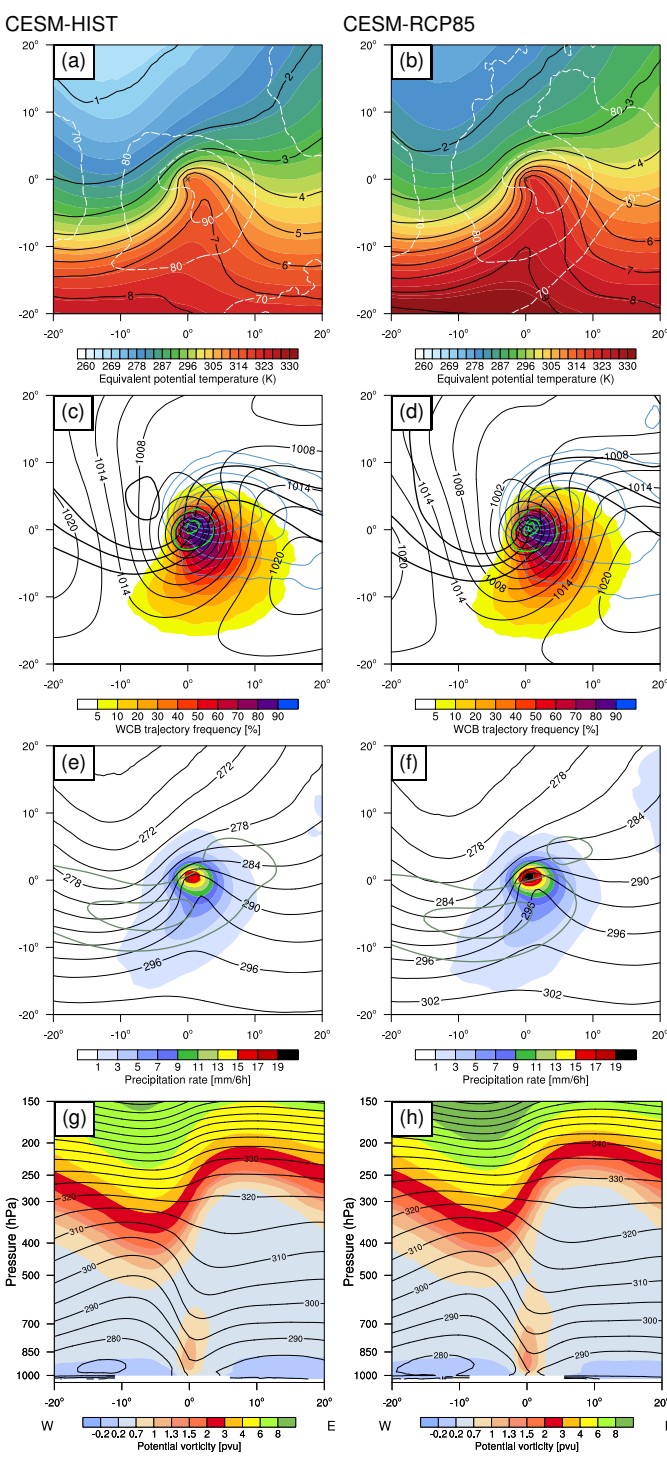

**Figure 7.** Composites of C1 cyclones in NH winter in (a, c, e, g) CESM-HIST and (b, d, f, h) CESM-RCP85 in the middle of the 24 h period of strongest deepening. (a, b) Equivalent potential temperature (K; shading), specific humidity (black contours every $g\,kg^{-1}$) and relative humidity (white dashed contours every 10%) at 850 hPa. Fields in (c, d) are as in Fig. 3a,b, fields in (e, f) as in Fig. 3c,d, and fields in (g, h) as in Fig. 3e,f.



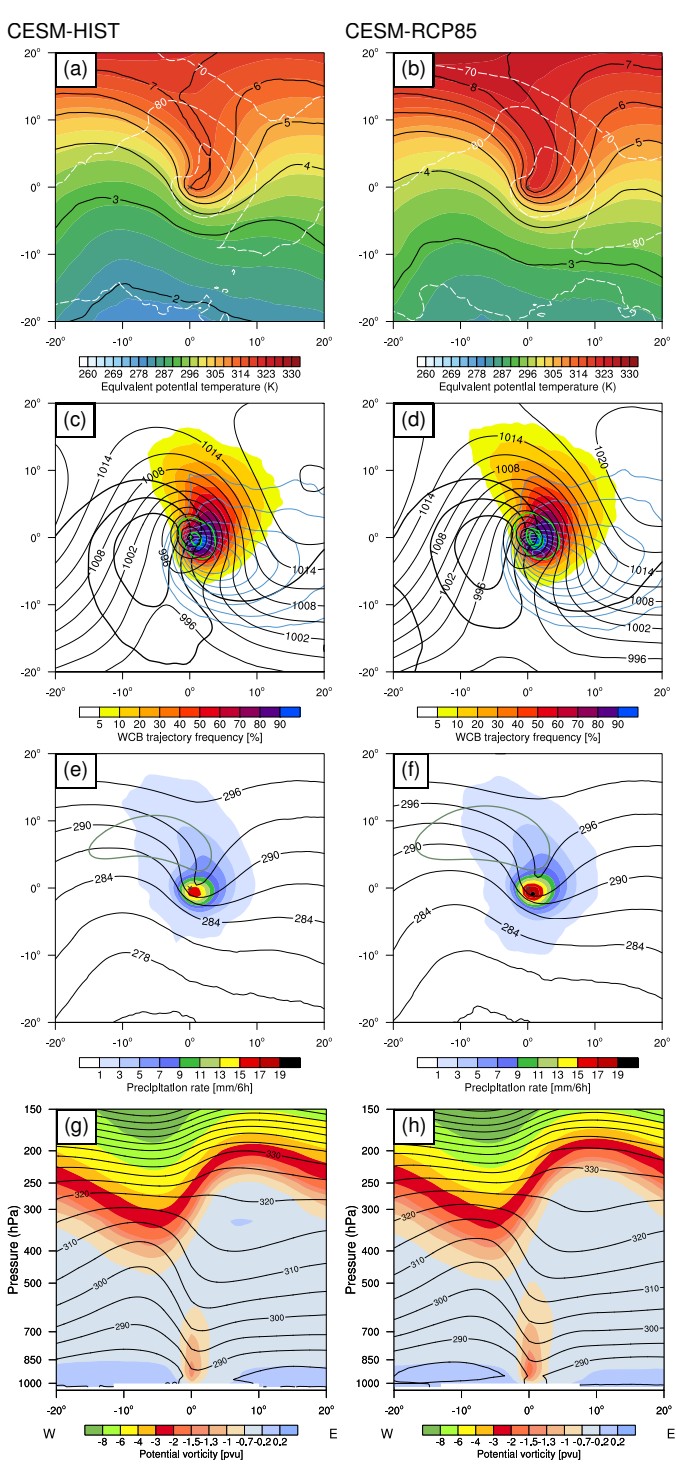

**Figure 8.** As Fig. 7, but for SH winter.





The low- and upper-level WCB frequencies have very similar magnitudes in CESM-HIST and CESM-RCP85 (Fig. 7c,d and 8c,d). This is in apparent contradiction to the previously discussed increase in the WCB strength in the future climate (Figs. 4c,d and 6). However, the low- and upper-level frequencies in these composites only indicate the fraction of C1 cyclones with at least one WCB air parcel at a specific grid point in the lower or upper troposphere – if a cyclone is associated with many WCB air parcels at a grid point, it does not contribute to higher frequencies in the composite than if the cyclone only has one WCB air parcel at this grid point. In CESM-RCP85, close to the cyclone centre the number of low- and upper-level WCB air parcels is typically higher, particularly in the SH (not shown). The percentage of C1 cyclones with trajectories producing high values of low-level PV increases around the cyclone centre by about 5% in the NH (Fig. 7c,d) and 10% in the SH (Fig. 8c,d) in the future simulations, pointing to an increase in the WCB-related cyclonic PV formation. In the same area, the maximum precipitation increases by $3\,\mathrm{mm\,h^{-1}}$ in both hemispheres (Fig. 7e,f and 8e,f) and the diabatically produced low-level PV intensifies by 0.3 pvu in the NH and 0.2 pvu in the SH and extends toward higher levels (Fig. 7g,h and 8g,h). The intensification of the PV anomaly and the extension toward higher levels are consistent with the enhanced diabatic heating rate along the WCB trajectories and the increase in the altitude of the maximum diabatic heating found in Joos et al. (2022). At upper levels, the cyclonically breaking Rossby wave and the jet have similar intensities in the present-day and future climate simulations (Fig. 7c-f and 8c-f).

Figure 9 shows the PV anomalies in a west-east vertical cross section through the centre of the composite cyclone. For each cyclone, before the compositing the PV anomalies have been calculated by subtracting the 50-year climatological mean PV value of the specific climate simulations from the actual PV value at this position. Consistent with the PV cross sections (Fig. 7g,h and 8g,h), in both hemispheres the diabatically produced cyclonic low-level PV anomaly amplifies in the future climate, i.e., it becomes more positive in the NH (Fig. 9a,b) and more negative in the SH (Fig. 9c,d). The upper-level disturbance to the west of the cyclone centre also corresponds to a cyclonic PV anomaly, and it intensifies as well in the future climate. However, the relative increase of the peak values is much larger for the cyclonic low-level PV anomaly than for the cyclonic upper-level PV anomaly (25.5% vs 11.1% in the NH and 14% vs. 7.2% in der SH, Tables 3 and 4). This suggests that the enhanced deepening and stronger intensity of the C1 cyclones in the future simulations observed in Fig. 6 is mainly associated with the enhanced WCB-related diabatic PV production, and in the SH additionally with the increased low-level baroclinicity. Downstream of the cyclone centre, the elevated tropopause goes along with anticyclonic upper-level PV anomalies, which also amplify in the future climate (Fig. 9 and Tables 3, 4). This amplification is consistent with the intensification of the WCB and the WCB-related latent heating, which results in a stronger transport of anticyclonic PV into the tropopause region (see introduction).

We also investigated the composite structure of the C1 cyclones 12 h later, at the end of the 24 h interval of explosive deepening (not shown). Also at this time step, the WCB-related diabatic PV production intensifies in the future climate (Fig. 6) and contributes to the formation of a stronger PV tower (not shown) and 1.7 hPa deeper minimum SLP in the NH and 2.5 hPa deeper minimum SLP in the SH, respectively (Fig. 6). The maximum values of the cyclonic low-level PV anomalies increase by almost 50% in the NH and 19% in the SH in the future climate, whereas the peak values of cyclonic upper-level PV only





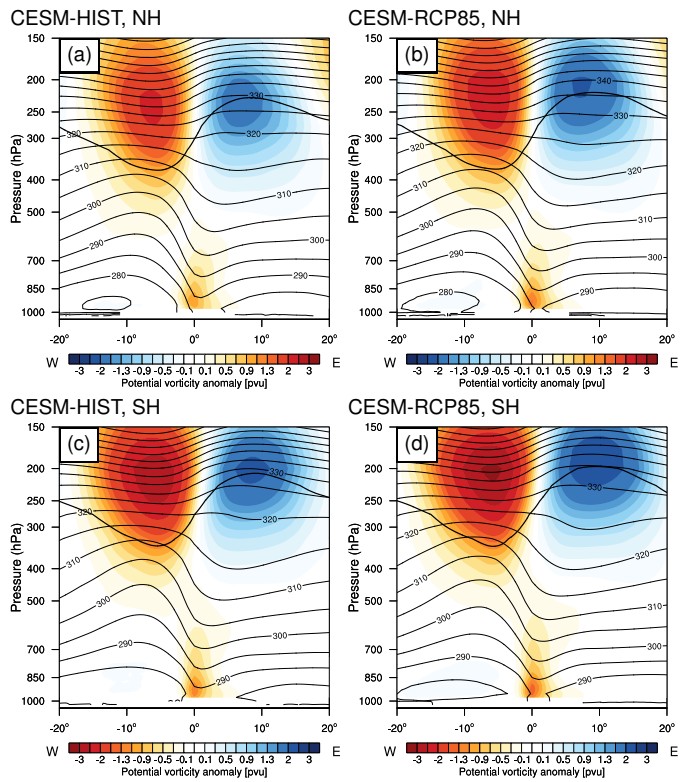

**Figure 9.** West-east oriented cross section across the centre of the composite cyclone, in the middle of the 24 h period of strongest deepening, showing the PV anomaly (pvu; shading), the 2 pvu contour (thick black contour) and potential temperature (thin black contours every 5 K). The fields are shown for (a, b) NH winter and (c, d) SH winter in (a, c) CESM-HIST and (b, d) CESM-RCP85.

**Table 3.** Maximum values of the cyclonic low-level, cyclonic upper-level and anticyclonic upper-level PV anomaly in C1 cyclones in NH winter in the middle ($t = 12$ h) and at the end ($t = 24$ h) of the 24 h period of strongest deepening in CESM-HIST and CESM-RCP85, and the relative change between the two simulation periods.

| NH | CESM-HIST | CESM-RCP85 | Relative change |
|---|---|---|---|
| Cyclonic low-level PV anomaly at $t = 12$ h | 1.06 pvu | 1.33 pvu | +25.5% |
| Cyclonic low-level PV anomaly at $t = 24$ h | 1.31 pvu | 1.96 pvu | +49.6% |
| Cyclonic upper-level PV anomaly at $t = 12$ h | 2.17 pvu | 2.41 pvu | +11.1% |
| Cyclonic upper-level PV anomaly at $t = 24$ h | 2.36 pvu | 2.64 pvu | +11.9% |
| Anticyclonic upper-level PV anomaly at $t = 12$ h | −2.10 pvu | −2.39 pvu | −13.8% |
| Anticyclonic upper-level PV anomaly at $t = 24$ h | −2.90 pvu | −3.17 pvu | −9.3% |





**Table 4.** Same as Table 3, but for SH winter.

| SH | CESM-HIST | CESM-RCP85 | Relative change |
|---|---|---|---|
| Cyclonic low-level PV anomaly at $t = 12\,\mathrm{h}$ | $-1.29\,\mathrm{pvu}$ | $-1.47\,\mathrm{pvu}$ | $-14.0\%$ |
| Cyclonic low-level PV anomaly at $t = 24\,\mathrm{h}$ | $-1.42\,\mathrm{pvu}$ | $-1.69\,\mathrm{pvu}$ | $-19.0\%$ |
| Cyclonic upper-level PV anomaly at $t = 12\,\mathrm{h}$ | $-2.92\,\mathrm{pvu}$ | $-3.13\,\mathrm{pvu}$ | $-7.2\%$ |
| Cyclonic upper-level PV anomaly at $t = 24\,\mathrm{h}$ | $-3.06\,\mathrm{pvu}$ | $-3.20\,\mathrm{pvu}$ | $-4.6\%$ |
| Anticyclonic upper-level PV anomaly at $t = 12\,\mathrm{h}$ | $2.40\,\mathrm{pvu}$ | $2.74\,\mathrm{pvu}$ | $+14.1\%$ |
| Anticyclonic upper-level PV anomaly at $t = 24\,\mathrm{h}$ | $3.11\,\mathrm{pvu}$ | $3.38\,\mathrm{pvu}$ | $+8.7\%$ |

increase by about 12% and 5%, respectively (Tables 3, 4). The anticyclonic upper-level PV anomalies downstream of the cyclone centre also increase (Tables 3, 4) and push the tropopause slightly further upward and toward the poles (not shown).

In summary, in both hemispheres it is projected that C1 cyclones will have even stronger WCBs, stronger WCB-related diabatic PV production, an extension of the diabatically produced PV toward higher levels and an increased precipitation rate
in the future climate. They will become warmer, moister and slightly more intense.

## 5  Summary and conclusions

In this study, we quantified the role of WCBs and their diabatically produced PV anomalies for cyclone intensification in present-day (1990-1999) and future (2091-2100) climate simulations of the Community Earth System Model Large Ensemble (CESM-LE). The present-day simulations (CESM-HIST) were based on historical climate forcing (Lamarque et al., 2010), and
the future simulations (CESM-RCP85) on the Representative Concentration Pathway 8.5 (RCP8.5) emission scenario (Lamarque et al., 2011; Meinshausen et al., 2011). The aims were to (i) assess whether the climate model is able to adequately represent the cyclone properties and the associated WCBs by comparing the present-day simulations with ERA-Interim reanalyses, and (ii) evaluate how climate change affects the importance of WCB-induced diabatic PV production for cyclone intensification in the winter season in both hemispheres. Such a detailed investigation of the role of WCBs for cyclone intensification in climate
models is not straightforward and was only possible because CESM-LE has been re-run to obtain 6-hourly three-dimensional output of the wind fields. To address the aims, a large number of cyclones and their associated WCB trajectories have been identified in the climate simulations and in ERA-Interim during NH and SH winter. Cyclone deepening has been measured by the maximum 24-h SLP change during the cyclone lifecycle, adjusted by latitude (Sanders and Gyakum, 1980), and the WCB intensity by the WCB air mass located in the lower troposphere ($p > 500\,\mathrm{hPa}$) during the 24 h interval of strongest in-
tensification of the associated cyclone. Based on the questions posed in the introduction, the key findings of the study can be summarised as follows:



1. Compared to ERA-Interim, the climate model is able to represent the properties and three-dimensional structure of extratropical cyclones, as well as the associated WCBs, remarkably well. Particularly in SH winter, there are very small differences between the present-day simulations and the reanalyses in terms of the cyclone deepening rates, the WCB strength and the statistical relationship between the two. In NH winter, the present-day simulations capture the deepening rates of the weak and medium-strong cyclones, but they underestimate them for the most explosive cyclones by 0.1-0.3 Bergeron. The WCB strength and the link between WCB strength and cyclone deepening rate is well captured also in the NH. In the subgroup of explosive cyclones with intense WCBs (C1), in both hemispheres the model is able to reproduce the composite fields of the cyclones during the deepening phase in terms of upper-level PV and jet structure, low-level potential temperature, SLP, the precipitation pattern and the position of the WCB trajectories, but it underestimates the diabatically produced low-level PV anomaly by about 0.5 pvu.

2. In the SH, comparison of the simulations reveals an increase in the WCB strength and – for the medium-strong and strongly intensifying cyclones – the cyclone intensification rate in the future climate. In the NH, the WCB strength is also projected to increase, but to a smaller extent than in the SH, and overall there are no significant changes in the cyclone deepening rates. The enhanced cyclone deepening rate of the medium-strong and strongly deepening cyclones in the SH but not in the NH could partly be associated with the stronger increase in the WCB strength and accordingly the WCB-related diabatic heating in the SH. Furthermore, it is consistent with the opposite changes in low-level baroclinicity projected in the two hemispheres with global warming (e.g., Harvey et al., 2014; Catto et al., 2019), i.e., an increase in the SH, which favourably interacts with the moist dynamics to create stronger storms, and a decrease in the NH, which counteracts the direct effects of the moist dynamics such that the storm strength does not change significantly.

3. In both hemispheres, the cyclone deepening rate correlates positively with the strength of the associated WCB, with a Spearman rank correlation coefficient in the present-day simulation of 0.68 in the NH and 0.51 in the SH, and in the future simulation of 0.66 in the NH and 0.55 in the SH, respectively. Thus, both in the present-day and future simulations, there is a distinct statistical signal that cyclones with a more intense WCB typically go along with stronger deepening. While many cyclones do not have any WCB, the majority of the explosive cyclones has strong WCBs. The number of explosive cyclones with a strong WCB (C1) increases in the future simulations, whereas the number of explosive cyclones with a weak WCB (C3) decreases. In addition to the increase in number, the C1 cyclones themselves are projected to be associated with even stronger WCBs, more WCB-related diabatic PV production in the cyclone centre, an extension of the diabatically produced PV toward higher levels, the formation of a stronger PV tower at the end of the explosive deepening phase, higher precipitation rates, and to become warmer and moister. Their deepening intensifies slightly, with an enhanced decrease in the minimum SLP by 2 hPa in the NH and 4 hPa in the SH at the time of the strongest intensity. Low-level baroclinicity in the cyclones' surroundings decreases slightly in the NH and increases slightly in the SH, consistent with the overall changes expected in the two hemispheres (e.g., Harvey et al., 2014). The relative increase in the diabatically produced PV anomaly is much stronger than the relative increase in the cyclonic upper-level PV anomaly upstream of the cyclone centre. This indicates that the stronger diabatic PV production in the WCBs, and





in the SH additionally the increased baroclinicity, are essential for the enhanced deepening and stronger intensity of C1 cyclones in a warmer climate.

The overall remarkably good representation of the cyclone structure and key properties in CESM-HIST in comparison to ERA-Interim is in line with the findings from Bengtsson et al. (2009) and Catto et al. (2010) for other climate models. However, the underestimation of the diabatically produced low-level PV anomaly by 0.5 pvu is relatively strong. A possible reason for this underestimation could be the lower spatial resolution (30 vertical levels in CESM-LE vs. 60 in ERA-Interim and about $1.25°$ vs. $1°$ horizontal resolution). Furthermore, the two models are based on different parameterization schemes for subgrid-scale processes, and in particular a different treatment of cloud microphysical and convective processes could contribute to the discrepancies in the intensity of the diabatically produced PV anomaly. Problems in the representation of cloud diabatic processes in extratropical cyclones have also been reported for other climate models (e.g., Govekar et al., 2014; Catto et al., 2015a; Hawcroft et al., 2016, 2017). However, also ERA-Interim is associated with biases in the representation of moist processes when compared to satellite data (Binder, 2017; Hawcroft et al., 2017). Aside from the fact that the two data sets are based on different models, an exact agreement cannot be expected because ERA-Interim only represents one possible realisation of the present-day climate, whereas CESM-HIST comprises an ensemble with equally likely realisations of the same decade. Altogether, the remarkably similar structure and properties of the extratropical cyclones and WCBs in the climate model and the reanalyses gives confidence in CESM-LE's projections for a warmer climate.

The increase in the WCB strength and in the number of C1 cyclones, as well as – for the C1 cyclones – the amplification of the WCB-related diabatic PV production, the increase in the precipitation rate and the slightly stronger deepening, all indicate that cyclones will become more diabatic in a warmer climate, and that WCBs will be even more important for explosive cyclone intensification. The enhancement of the cyclone-related precipitation and of the diabatically produced PV anomaly are in agreement with previous results based on climate models (Bengtsson et al., 2009; Hawcroft et al., 2018; Dolores-Tesillos et al., 2022) and idealised simulations (Pfahl et al., 2015; Büeler and Pfahl, 2019; Sinclair et al., 2020).

In CMIP6 models, Priestley and Catto (2022) reported for both hemispheres a future increase in the intensity of the most extreme wintertime extratropical cyclones, as measured in terms of peak low-level vorticity. In our simulations, in the SH strongly deepening cyclones in general and in the NH those with intense WCBs also become slightly deeper. However, in the NH this enhancement of the deepening in C1 cyclones is relatively small, despite the concomitant increase in the WCB-related diabatic heating. This confirms the findings from previous studies (see, e.g., the reviews by Shaw et al., 2016; Catto et al., 2019) that the impact of rising temperatures and enhanced cloud-diabatic processes on cyclone intensity is not straightforward, as the increase in the diabatic heating is embedded in various other, partly compensating processes like changes in baroclinicity, vertical stability and the tropopause height.

A limitation of this study is that the results are based on one single climate model, and that the model has relatively coarse horizontal resolution ($\sim 1°$). Nevertheless, the availability of three-dimensional model level output at high temporal resolution enabled unprecedented insight into the Lagrangian evolution of cyclones and their three-dimensional PV structure in a climate model, and thereby provided a better process understanding of cyclone intensification in a future climate.



*Data availability.*  The ERA-Interim reanalyses can be downloaded from the ECMWF web page (https://apps.ecmwf.int/datasets/data/interim-full-daily/levtype=sfc/, last access: July 2022). The CESM source code that was used for the CESM-LE simulations is available from https://www.cesm.ucar.edu/models/cesm1.0/ (last access: July 2022). The model output of the CESM-LE reruns and the WCB and cyclone data used in this study are available from the authors upon request.

*Author contributions.*  HB designed and performed this study, using WCB trajectories previously identified by MS and HJ. HB wrote the
manuscript, with feedback about the results and text from all co-authors.

*Competing interests.*  One of the (co-)authors is a member of the editorial board of Weather and Climate Dynamics. The authors have no other competing interests to declare.

*Acknowledgements.*  We thank Urs Beyerle (ETH Zurich) for performing the CESM-LE reruns, and MeteoSwiss and ECMWF for granting access to the ERA-Interim reanalyses. We are grateful to Dominik Büeler (ETH Zurich) and Matthias Röthlisberger (ETH Zurich) for
valuable comments and discussions. HB received funding from the Swiss National Science Foundation (project 185049) and from the European Research Council H2020 research and innovation program (INTEXseas, grant no. 787652).



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
