# Peer review of "Warm conveyor belts in present-day and future climate simulations. Part II: Role of potential vorticity production for cyclone intensification"

_Weather and Climate Dynamics, 2022_

## Referee Comment (RC3)

**Review of "Warm conveyor belts and future climate simulations. Part II: Role of potential vorticity production for cyclone intensification" by H. Binder, H. Joos, M. Sprenger & H. Wernli.**

This study aims to investigates how climate change affects WCB strength and the relation between WCB strength and cyclone deepening rate by comparing two climate model simulations which have been re-run with the Community Earth System Model Large Ensemble. The main result is that in the SH both the WCB strength and the cyclone deepening rate increase whereas in the northern hemisphere the WCB strength increases slightly but there are no changes to cyclone deepening rates. The difference between the hemispheres is convincingly explained by the differing responses of low-level baroclinicity to climate change. Overall, the manuscript is interesting, well within the scope of the journal, and clearly written. I have two major comments (details below) that the authors need to carefully consider before this manuscript can be accepted. In addition, I have lots of minor comments which I hope the authors find useful and can be used to improve the manuscript.

**Major Comments**

1. **Metric to quantify the WCB strength.**

   (a) I appreciate that this metric has been used before but I am not convinced that the number of trajectories accurately tells us how "strong" a warm conveyor belt is for two main reasons: (1) I would expect that a WCB that ascends faster (larger vertical motion) is stronger (more mass flux through a given level), but the current metric does not account for this – trajectories are all assumed to be a WCB as long as they meet the threshold and (2) defining the WCB strength by counting the number of trajectories also means that the geographical size of the cyclone / warm sector can influence the "strength" of the WCB. Point (2) is worthy of careful consideration since some studies have suggested that extra-tropical cyclones, (at least their wind field) is likely to increase in size in the future. Even if these concerns have been considered in previous studies, they should also be included here – there are very few details of this metric presented in this study and adding more information could hopefully remove my concerns about the applicability of this metric.

   (b) The definition of a WCB trajectory – trajectories must ascend 600 hPa within 48 hours. Again, I appreciate that this diagnostic has been used extensively before but mainly in a weather / current climate setting. Now it is used in a future (warmer) climate, I am not sure using a fixed threshold (600 hPa) is valid as the tropopause will be higher in the warmer climate and as such a trajectory needs to ascend a smaller fraction of the troposphere in the future to be a WCB than in the current climate. Can you convince a reader the fixed threshold is still appropriate?

2. **Link between WCB strength and how diabatic a cyclone is**

   (a) In the abstract it is stated "cyclones will be more diabatic in a warmer climate" and again in lines 328-329 where it is stated that a stronger WCB means that cyclones are more diabatic. This is a very strong statement given the evidence presented in this manuscript as I don't think this is such a simple step to go from stronger WCB to more diabatic. For example, the ascent in the WCB is largely driven by warm air advection and usually

stronger warm air advection occurs where there is a stronger north-south temperature gradient and more baroclinicity. Therefore, just because there is a stronger WCB does not mean that the relative forcing (between dry dynamics and diabatic processes) changes. To really claim robustly that cyclones are becoming more diabatic, the diabatic temperature tendencies would need to be computed.

**Minor Comments**

1. Section 2.3: It is not clear why a different approach for assigning WCB trajectories to the cyclones is taken in CESM-LE compared to in ERA-Interim. Please revised the manuscript to justify why this is the case.

2. Section 2.3, lines 176 - 177. It is not completely clear how ΔSLP is computed. Is it computed every 6 hours and over a 24-hour period or only computed every 24 hours, over a 24-hour period? Please clarify.

3. Section 3: Why compare to ERA-Interim which is now quite old? ERA5 has been available for a considerable time now and it would have more appropriate to compare to ERA5. Given the much higher resolution of ERA5 compared to ERA-Interim, I think the low-level PV anomaly associated with WCB in ERA5 is likely larger than in ERA-Interim which would mean the difference between ERA5 and CESM-LE might be very large.

4. Lines 211. "about half of the cyclones do not have a WCB in ERA-Interim and CESM-HIST". This sentence really confused me to start with and as such I think it is misleading. Those cyclones probably do have a warm, ascending airstream (i.e. a weak or shallow WCB) associated with them but it does not meet the requirements to be defined as a WCB here. This sentence needs to be revised. This comment also applies to line 445.

5. Line 222. Why are the correlations between WCB strength and cyclone deepening rate smaller in the southern hemisphere compared to the northern hemisphere in both ERA-Interim and CESM? This is interesting – is it because SH cyclones are more driven by the low-level baroclinity and less by diabatic processes than in the NH?

6. Line 253, 376, and elsewhere. Units of precipitation. Here in the text it is written mm/h, but in Figure 3 it is mm / 6hr. Please check these. Also the precipitation rates (even if mm/6hr) seem to be a little larger than I would expect in a composite as usually on weather maps of day-to-day maximum values of precipitation in one mid-latitude cyclone are around ~5 mm / h. I'd expect the compositing to average / smooth things out leading to smaller values.

7. Line 269 and line 433 "remarkably well". This is too strong a statement – it would be more accurate to say there is reasonable agreement. The difference of 0.5 PVU in the intensity of the low-level PV anomaly is, in percentage terms, quite large.

8. Line 278. Is the difference in the number of cyclones statistically significant? This could be tested using the numer of cyclone each winter and comparing the two populations.

9. Line 347. Suggest change "higher" to "larger" as I first though that the WCB peak was moving upwards in the atmosphere.

10. Line 382 – 395, Figure 9. Does the vertical tilt change between CESM-HIST and CESM-RCP85? I think this analysis would be a small but valuable addition to this manuscript since some studies (that you cite in the introduction) have shown that the low-level PV anomaly

moves downstream which limits the coupling and interaction between the low level and upper level anomaly.

**Figures and Tables**

1. Figure 2. Suggest moving the C1 and C2 labels to a part of the "box" where there is no data as they are hard to see. Also the arrow pointing to the C3 part of the phase space is very easy to miss – can it be made more obvious?

2. Figure 3. Check the units of precipitation (see minor comment #6 above).  Add how many cyclones are included in these composites.

3. Figure 4. When printed, it is very difficult to see the grey shading. Can it be made darker or the edges shown by dashed lines?

4. Figure 5. This shows the mean number of each type of cyclone per winter. Could some range of the variability be added to this figure (related to minor comment #8)

5. Figure 6. Add to the caption what t=0 corrasponds to.

6. Figure 7 & 8. Add how many cyclones are included in these composites either to the title (after CESM-HIST etc.) or in the caption.

---

## Author Response (AR1)

*Paper wcd-2022-37*

**Warm conveyor belts in present-day and future climate simulations. Part II: Role of potential vorticity production for cyclone intensification**

by Hanin Binder, Hanna Joos, Michael Sprenger, and Heini Wernli

*Reply document*

We thank all three reviewers for their thoughtful and constructive comments that helped us to improve the manuscript. Based on the reviewers' suggestions, we implemented several changes in the manuscript. The main changes are that we:

- more carefully formulate our main conclusions (in response to comments by reviewers 2 and 3),
- better explain and motivate our choice for the WCB intensity metric (in response to comment by reviewer 3),
- now include more references about earlier studies that have focused on GCM evaluation of extratropical cyclones (suggestion by reviewer 2),
- and improved most figures according to the suggestions by reviewers 2 and 3.

Below we provide a one-to-one response to all points raised by the reviewers. The reviewers' comments are in black and our replies in blue.

**Reviewer 1 (Derek J. Posselt)**

This paper describes an analysis of cyclones and WCBs in the ERA-Interim analysis, compared with output from an ensemble of climate model simulations in present day and future climate conditions. The authors use established storm and WCB identification techniques to examine how storms and WCBs may change in future climates, with an eye toward the role of diabatically produced potential vorticity. They conclude that the climate model produces a realistic spectrum of storms and WCBs, relative to reanalysis. There are increases in WCB strength, especially in the southern hemisphere, and these increases may be related (in the SH specifically) to increases in the strength of the most intense storms. In both hemispheres, increased WCB strength correlates with increased storm deepening rates.

I found this paper to be well written, and the analyses well conceived. The research is a natural extension of the already impressive body of work conducted by the authors on this topic, and is an important contribution to our understanding of how extreme weather events may change in future climate states.

Many thanks for this positive assessment!

I have only a one minor comment for the authors to consider.

In a follow-up to their 2018 paper, Tierney et al. (2019) indicated that there were potential relationships between the non-monotonic response of storm EKE and the presence of convection (and the potential effect on PV phase locking). I wonder what effect changes in convection in future climates might have on the WCBs and storm intensity and intensification presented here? The authors do note that convective parameterization differences make comparison between reanalysis and climate models challenging (and the parameterizations themselves make analysis of convection difficult), but perhaps they could comment on the possible role of convection in future ETC / WCB changes?

References:
Tierney, G. T., D. J. Posselt, and J. F. Booth, 2019: The Impact of Coriolis Approximations on the Environmental Sensitivity of Idealized Extratropical Cyclones. Clim. Dyn., 53, 7065-7080. https://doi.org/10.1007/s00382-019-04976-x

This is a very relevant (and difficult) question. Embedded convection indeed plays an important role in many extratropical cyclones and their WCBs, and it can significantly alter the diabatic PV modification in WCBs, in particular in the mid and upper troposphere where vertical wind shear is typically largest (e.g., Oertel et al. 2020, 2021). From these studies, our current understanding is that embedded convection can influence the upper-level jet and downstream ridge formation, however, we are unsure about its more local and low-level effects on the cyclone itself. We therefore don't dare to speculate about the effects of changes in convection in future climates on storm intensity. But we now mention this open question in the discussion of our results, as well as the need to eventually study future storm changes and the role of diabatic processes also in simulations with much higher spatial resolution that partially resolve convective processes, see L489-496 in the revised manuscript.

Oertel, A., M. Boettcher, H. Joos, M. Sprenger, and H. Wernli, 2020. Potential vorticity structure of embedded convection in a warm conveyor belt and its relevance for large-scale dynamics. Weather Clim. Dynam., 1, 127–153.

Oertel, A., M. Sprenger, H. Joos, M. Boettcher, H. Konow, M. Hagen, and H. Wernli, 2021. Observations and simulation of intense convection embedded in a warm conveyor belt – how ambient vertical wind shear determines the dynamical impact. Weather Clim. Dynam., 2, 89–110.

**Reviewer 2**

Brief Summary: This manuscript analyzes the ability of CESM-LE to represent warm conveyor belts (WCB) as compared to reanalysis. Then the paper examines how WCBs are forecast to change in the future in the CESM-LE.

Overall impression: the paper is well-written in terms of clarity, grammar and intent. I appreciate that the authors lay out their research questions in the introduction and then return

to them in the conclusion. I think the methodology is sound and do not see a need to a significant amount of additional work.

Thank you for this overall positive impression of our study!

However, I have some comments about the interpretation of the results and a few questions that will require some extra analysis.

Major Comment
The authors conclude that there is a clear signal of an increase in WCB strength and enhanced WCB-related diabatic heating in the SH. I agree with this to some extent, but I would be a bit more cautious in how I would state the results – both in the abstract and in the conclusion section. The reason I say this is because, based on Fig. 3e vs. Fig. 3f, the difference in low-level PV in the core of the ETC composite for ERA-Interim vs CESM-Hist is of a similar magnitude as the difference between CESM-Hist and CESM-RCP85 (i.e., Fig. 7g,h and Fig. 8g,h). This means, if we accept reanalysis as truth, the bias in the model for present day is of a similar magnitude as the projected change in low-level PV. While I agree that some of the other variables analyzed do not show the same sort of issue – e.g., the climate change signal is larger than the model bias for precipitation, if it were me writing the paper, I would be more cautious in how I deliver the take home message about the modeled projections of the WCB and associated diabatically generated PV. Thus, I expect the authors to either add more explanation as to why such caveats are unnecessary, or adjust the language in the abstract and the conclusions to illustrate the amount of uncertainty that the figures appear to show.

Thank you, this is a good point. We now mention the fact about the model bias in the present-day climate in the abstract (L10 in the revised manuscript). This bias was already mentioned in the conclusions (L442 in the revised manuscript), but we added a note of caution also when discussing the stronger diabatic PV production in the WCBs in the future climate (L466 in the revised manuscript). We think that a main reason for this bias is model resolution. Whereas ERA-Interim has a horizontal resolution of 80 km and 60 vertical levels, the resolution of CESM-LE is slightly coarser with about 100 km horizontal grid spacing and only 30 vertical levels. When considering diabatic PV production, then vertical resolution matters mainly to capture strong vertical gradients in latent heating and horizontal resolution for peaks in relative vorticity (note that, to first order, diabatic PV production is proportional to the vertical heating gradient times the vertical component of absolute vorticity).

Minor Comments
L 55-80: Somewhere in the introduction, perhaps in this section (L 55–80), I think it would make sense to refer some of the studies that have focused on GCM evaluation of extratropical cyclones for the processes and mechanisms that the authors are focused on. For instance, the Catto et al. (2010) work on ETCs in general; the Hawcroft et al. (2015) and Booth et al. (2018) work for ETC precip, and the Riviere et al. (2021) work that focuses directly on the WCB in a global model. My feeling is: discussing these works in the introduction sets context for the GCM evaluation analysis that you will do and shows some of the successes of the

models. At the same time, we can't trust the models completely, especially in the southern hemisphere, see for instance, Chemke et al., (2022).

We apologize for this oversight, and we thank the reviewer for these suggestions. We now include references to the first three papers in L57 and to Chemke et al. in the final discussion on L506. The paper by Rivière et al. is not directly relevant for this study, as it does not discuss the effect of the WCB on cyclone intensification.

Catto, J. L., L. C. Shaffrey and K. I. Hodges, 2010: Can Climate Models Capture the Structure of Extratropical Cyclones? J Climate, 23, 1621–1635.

Hawcroft M. K., Shaffrey L. C., Hodges K. I., Dacre H. F., 2015: Can climate models represent the precipitation associated with extratropical cyclones? Climate Dynamics. 1–17. doi:10.1007/s00382-015-2863-z

Booth J. F., C. M. Naud, J. Willison, 2018: Evaluation of Extratropical Cyclone Precipitation in the North Atlantic Basin: An analysis of ERA-Interim, WRF, and two CMIP5 models. J Climate, 31:6, 2345-2360.

Rivière, G., Wimmer, M., Arbogast, P., Piriou, J.-M., Delanoë, J., Labadie, C., Cazenave, Q., and Pelon, J.: The impact of deep convection representation in a global atmospheric model on the warm conveyor belt and jet stream during NAWDEX IOP6, Weather Clim. Dynam., 2, 1011–1031, https://doi.org/10.5194/wcd-2-1011-2021, 2021

Chemke, R., Ming, Y. & Yuval, J. The intensification of winter mid-latitude storm tracks in the Southern Hemisphere. Nature Climate Change 12, 553–557 (2022). https://doi.org/10.1038/s41558-022-01368-8

L138: You write: "In total, this yields 50 simulated years for each time period." I interpret this to mean that you are not looking at the data from the different ensemble members collectively, is that correct? I am used to ensembles being used for inter-comparison, but here you use them simply to have more data. That is fine, but a sentence clarifying that would be helpful.

Yes, we look at data from the 5 ensemble members collectively, and, as you write, we use the ensemble as a means to have more data, which enables statistically more robust analyses. We clarify this in the revised manuscript in L142.

L142: Using 1979–2014 for reanalysis as compared to 1990–1999 for the GCM might be an issue, but perhaps not? There are differences in ENSO variability for the two time periods which might influence midlatitude mean state and thereby influence the WCBs and ETCs. Or perhaps not. You discuss this later on, in the section where you discuss the figure, but I suggest moving or adding something explanation right here, where you introduce the models as having been analyzed for different epochs.

Indeed, this comparison is not perfect, but since the CESM-LE simulations are coupled atmosphere-ocean simulations, they anyway do not reproduce the observed ENSO variability. Therefore, limiting the comparison to 1990-1999 for the reanalysis would not make the comparison better in terms of, e.g., ENSO variability, but it would make the results statistically less reliable. We think that our approach is pragmatic and meaningful, but we agree that it's a good idea to mention the different time periods and ENSO variabilities when describing the data (see L150).

L165-166: Why are different methods used for assigning WCB trajectories in ERAi vs CESM? This seems like a possibly crucial issue given how closely you are comparing the results garnered from this analysis. I think the reader would benefit from some explanation of this choice, and some reassurance as to why it should not be an issue.

We acknowledge that this is a slight weakness in the design of our study, which we spotted rather late. Luckily, this small methodological difference is not an issue for the comparison of ERA-Interim with CESM-HIST. We made a comparison of the two methods for 10 simulated years in CESM-HIST (whereas 50 years are used in the paper) and find only very small differences in the percentile curves of the WCB strength associated with the cyclones (see Fig. R1 below). Given these negligible differences, and the substantial computational time required to redo our evaluations, we decided to keep the small methodological difference, which do not affect the main conclusions of our study.

[Figure]

Figure R1. Percentile curves of the WCB strength associated with the cyclones in NH winter in CESM-HIST (as Fig. 1c in the paper). Here the curves show the results when using the two slightly different methods to attribute WCB trajectories to cyclones. The curves are noisier than in Fig. 1 in the paper because here only 10 years were analyzed instead of 50 years. Results for the SH winter are qualitatively very similar (not shown).

Line 229: Table 1: It is a bit of a surprise to me that there are more cyclones in the NH than in the SH. Are the SH events longer-lived? There is more storminess in the SH than the NH isn't there? A comparison of the Hoskins and Hodges 2002 and 2005 papers suggests that there is more in the SH. The lack of land masses down there also seems to suggest that there would be more cyclone activity over the Southern Ocean as compared to the NH. Why do you think you've found a different result?

Hoskins, B. and K. Hodges, 2002: New Perspectives on the Northern Hemisphere Winter Storm Tracks. J. of Atmos. Sci., 59, 1041-1061.

Hoskins B. and K. I. Hodges,, 2005: A new perspective on Southern Hemisphere storm tracks. J. Climate, 18, 4108-4129.

The number of cyclones is a tricky quantity. Different cyclone tracking methods do not agree well in terms of cyclone numbers (Raible et al., 2008; Neu et al., 2013), and this is mainly related to a different treatment of weak cyclones and of splits and mergers of cyclone tracks. Our algorithm (M20 in Neu et al., 2013) finds substantially more cyclones in the NH than in the SH, both in winter and summer (see Tables 2 and 3 in Neu et al., 2013), most likely because of more frequent continental heat lows in the NH, and because of more frequent splits of cyclone tracks near topography. From the 15 cyclone tracking methods that participated in the Neu et al. comparison project, 13 methods showed the same behavior (more tracks in NH than in SH). Since the Hodges method did not participate in the Neu et al. study, a direct comparison is difficult. Our hypothesis is that the longer lifetime threshold of 2 days used by Hoskins and Hodges, and their T42 truncation of the input fields eliminate some of the shorter-lived and smaller-scale cyclones, which are more frequent in the NH than in the SH.

Neu, U., et al., 2013. IMILAST: A community effort to intercompare extratropical cyclone detection and tracking algorithms. Bull. Amer. Meteor. Soc., 94, 529–547.

Raible, C. C., P. M. Della-Marta, C. Schwierz, H. Wernli, and R. Blender, 2008. Northern Hemisphere extratropical cyclones: A comparison of detection and tracking methods and different reanalyses. Mon. Wea. Rev., 136, 880–897.

Figure 2: In panel b, the maximum on the y-axis is different from the other panels. Also, the number values shown in the y-axis are a bit non-traditional. Is there a reason for that?

Thank you, the figure has been improved.

Figure 2 suggests that the probability of a cyclone reaching bomb strength without much WCB air mass is large. Why do you think that is?

Indeed, in the so-called category C3 there are many strongly intensifying cyclones with almost no WCB. We discussed this phenomenon in some detail in Binder et al. (2016), where we concluded that "The category of explosively intensifying cyclones with weak WCBs is

inhomogeneous but often characterized by a very low tropopause or latent heating independent of WCBs.".

Figure 5: The text on the x-axis looks to have been cut-off at the bottom, e.g., the word Bombs is cropped too much.

Thank you, the figure has been improved.

Figure 5: In terms of the changes in the cyclone characteristics that relate to SLP. I just wonder if the normalization to a fixed latitude doesn't do enough for the southern hemisphere, where the gradient of the zonal mean of SLP with respect to latitude is very large in some locations. A plot that I would like to see is this:
- A histogram of the latitudes of the cyclone centers in the HIST and the RCP8.5 run on the same plot.
- A plot of the zonal mean of the climatology of the SLP for HIST and RCP8.5

My question on this is because I wonder how much a latitudinal shift in the location of the cyclones, or a change in the SLP climatology impacts a metric like the Bergeron.

Thanks for this suggestion. We produced a histogram of the latitudes of the cyclone centers in the HIST and the RCP85 simulations on the same plot (Fig. R2). For each cyclone, the latitude was determined in the middle of the 24-h period of maximum intensification. Our interpretation of the figures is that there is no obvious change between HIST and RCP85 in terms of latitude of maximum intensification, and therefore this should not affect the Bergeron metric.

[Figure]

Figure R2: Histograms of latitude of maximum cyclone intensification in CESM-HIST and CESM-RCP85 for (left) the NH and (right) the SH.

L309: In the introduction, and throughout the paper there are multiple references to a projected increase in baroclinicity in the SH, and all of the references point to a single paper. Given the data at the authors disposal, I wonder if it would make the paper stronger if the authors also calculate the change in baroclinicity and include that figure in the manuscript?

Thanks also for this valuable suggestion. In our original manuscript, similarly to Catto et al. (2019), we referenced Harvey et al. (2014) about the baroclinicity increase in the SH. We now calculated the Eady growth rate (EGR) in CESM-HIST and its change in CESM-RCP85 – which also considers changes in static stability – and this measure shows a more complex picture with mainly a decrease of EGR also in the SH (see Fig. R3). We suggest that more detailed analyses are required to understand this qualitative disagreement with Harvey et al. (2014) and we more carefully write about potential climate change effects on baroclinicity in the SH. More specifically, we reduced the references to the results by Harvey et al. (2014) about an increase of baroclinicity in the SH and a decrease in the NH, respectively.

It should also be noted that Harvey et al. (2014) considered a zonal-mean temperature difference between a tropical and polar reservoir at 850 hPa, whereas Fig. R3 shows large spatial variability of EGR also within the extratropics. And, also most likely important, the mean meridional temperature contrast (or EGR) might not be directly relevant for individual cyclones (e.g., of type C1) because the environment of these cyclones might deviate strongly from time-mean conditions. Therefore, our composites of low-level temperature associated with C1 cyclones (Fig. 7e,f; Fig. 8e,f) are particularly relevant, and they show a relatively clear reduction of the 850-hPa temperature gradient in the NH and almost no change in the SH, which is consistent with the fact that in the two hemispheres cyclone intensification responds differently to an increase in WCB strength.

[Figure]

Figure R3: Difference of Eady growth rate between CESM-RCP85 and CESM-HIST in (left) DJF and (right) JJA.

**Reviewer 3**

This study aims to investigates how climate change affects WCB strength and the relation between WCB strength and cyclone deepening rate by comparing two climate model simulations which have been re-run with the Community Earth System Model Large Ensemble. The main result is that in the SH both the WCB strength and the cyclone deepening rate increase whereas in the northern hemisphere the WCB strength increases slightly but there are no changes to cyclone deepening rates. The difference between the hemispheres is convincingly explained by the differing responses of low-level baroclinicity to

climate change. Overall, the manuscript is interesting, well within the scope of the journal, and clearly written.

Thank you for this positive statement!

I have two major comments (details below) that the authors need to carefully consider before this manuscript can be accepted. In addition, I have lots of minor comments which I hope the authors find useful and can be used to improve the manuscript.

Major Comments
1. Metric to quantify the WCB strength.
(a) I appreciate that this metric has been used before but I am not convinced that the number of trajectories accurately tells us how "strong" a warm conveyor belt is for two main reasons: (1) I would expect that a WCB that ascends faster (larger vertical motion) is stronger (more mass flux through a given level), but the current metric does not account for this – trajectories are all assumed to be a WCB as long as they meet the threshold and (2) defining the WCB strength by counting the number of trajectories also means that the geographical size of the cyclone / warm sector can influence the "strength" of the WCB. Point (2) is worthy of careful consideration since some studies have suggested that extratropical cyclones, (at least their wind field) is likely to increase in size in the future. Even if these concerns have been considered in previous studies, they should also be included here – there are very few details of this metric presented in this study and adding more information could hopefully remove my concerns about the applicability of this metric.

Thank you, and we agree that one can think of several meaningful ways of defining "WCB strength". This comment addresses two aspects:

- The reviewer suggests that the strength of a WCB should be measured by vertical velocity. We don't immediately understand why this should be more relevant than our measure for WCB strength. We are interested in the total vertical mass transport that reaches into the upper troposphere, where WCBs can induce negative PV anomalies that interact with the jet / isentropic PV gradient. Vertical velocity at a certain level is most likely not a good indicator for this.
- The reviewer mentions that, with our definition, the size of a cyclone and/or warm sector can influence the strength of the WCB. We agree and we think that this is perfectly fine. A larger cyclone with a large warm sector might in general transport more boundary layer air to the upper troposphere than a smaller cyclone, and this is what we want to quantify with our WCB strength.

Maybe it is helpful to add one important information about our trajectory setup: When calculating the WCB climatologies (in ERA-Interim and in CESM), we launch the 2-day forward trajectories on a regular grid with $\Delta x = 80$ km horizontal and $\Delta p = 20$ hPa vertical spacing between 1050 and 790 hPa. With this setup, every trajectory represents the same mass, given by $\Delta m = g^{-1} (\Delta x)^2 \Delta p$, and therefore the number of trajectories is directly proportional to the Lagrangian mass transport that exceeds the 600 hPa threshold. We now better motivate our measure for WCB strength by adding "We regard this as a useful measure

of WCB strength, as the number of trajectories is directly proportional to the Lagrangian mass transport that exceeds the 600 hPa threshold" (see L191).

(b) The definition of a WCB trajectory – trajectories must ascend 600 hPa within 48 hours. Again, I appreciate that this diagnostic has been used extensively before but mainly in a weather / current climate setting. Now it is used in a future (warmer) climate, I am not sure using a fixed threshold (600 hPa) is valid as the tropopause will be higher in the warmer climate and as such a trajectory needs to ascend a smaller fraction of the troposphere in the future to be a WCB than in the current climate. Can you convince a reader the fixed threshold is still appropriate?

Thanks for this question. We agree that using a fixed threshold has always some limitations. For instance, with our standard threshold of 600 hPa in 48 hPa, it is almost impossible to identify WCBs at high latitudes where the tropopause is lower. Eckhardt et al. (2004), for their WCB climatology, used a latitude dependent threshold ("60% of the zonally and climatologically averaged tropopause height at the trajectory's latitudinal position after 2 days"). Thinking along this line, one could adjust the 600 hPa criterion by considering the change in a warmer climate of the averaged tropopause height. However, in our view, this would make this, to the best of our knowledge, first Lagrangian analysis of WCBs in climate model simulations much more difficult to interpret. And estimating the relevant change in tropopause height would not be straightforward, since WCBs occur mainly in specific regions such that global mean tropopause height changes might not be appropriate. In the revised manuscript, we briefly mention the aspect that tropopause height is increased in the warmer climate (this aspect is discussed in more detail in the Part 1 paper by Joos et al.), but we prefer to keep the 600 hPa ascent criterion for both climate periods. We added the following sentence in LXX: "Note that we decided to use the same threshold for ascent to identify WCB trajectories in both climate periods, despite the fact that the extratropical tropopause rises slightly in the warmer climate (Joos et al., 2022).".

2. Link between WCB strength and how diabatic a cyclone is
In the abstract it is stated "cyclones will be more diabatic in a warmer climate" and again in lines 328-329 where it is stated that a stronger WCB means that cyclones are more diabatic. This is a very strong statement given the evidence presented in this manuscript as I don't think this is such a simple step to go from stronger WCB to more diabatic. For example, the ascent in the WCB is largely driven by warm air advection and usually stronger warm air advection occurs where there is a stronger north-south temperature gradient and more baroclinicity. Therefore, just because there is a stronger WCB does not mean that the relative forcing (between dry dynamics and diabatic processes) changes. To really claim robustly that cyclones are becoming more diabatic, the diabatic temperature tendencies would need to be computed.

Thank you for this very good suggestion. We now use a better wording in the abstract and in Sect. 4.1. What we meant by "cyclones will be more diabatic" is "latent heating associated with WCBs (as identified with our method) will increase". We did not intend to make a statement about the relative role of baroclinic vs. diabatic forcing.

Minor Comments

1. Section 2.3: It is not clear why a different approach for assigning WCB trajectories to the cyclones is taken in CESM-LE compared to in ERA-Interim. Please revised the manuscript to justify why this is the case.

See reply to comment about L165-166 from Reviewer 2.

2. Section 2.3, lines 176-177. It is not completely clear how ΔSLP is computed. Is it computed every 6 hours and over a 24-hour period or only computed every 24 hours, over a 24-hour period? Please clarify.

ΔSLP is computed every 6 hours for the next 24-hour period. We clarified this in the revised manuscript (see L186).

3. Section 3: Why compare to ERA-Interim which is now quite old? ERA5 has been available for a considerable time now and it would have more appropriate to compare to ERA5. Given the much higher resolution of ERA5 compared to ERA-Interim, I think the low-level PV anomaly associated with WCB in ERA5 is likely larger than in ERA-Interim which would mean the difference between ERA5 and CESM-LE might be very large.

Indeed, ERA5 has a higher spatial and temporal resolution than ERA-Interim and, most likely, represents the "true state" of the atmosphere even better. The main reason why we use ERA-Interim is that it enables a fairer comparison with CESM-LE. If we used ERA5 for comparison then, as the reviewer points out, we might get more substantial differences (e.g., in low-level PV) and it would be even more difficult to estimate whether differences between CESM and the reanalyses are an issue of the climate model or "simply" an effect of model resolution.

4. Lines 211. "about half of the cyclones do not have a WCB in ERA-Interim and CESM-HIST". This sentence really confused me to start with and as such I think it is misleading. Those cyclones probably do have a warm, ascending airstream (i.e. a weak or shallow WCB) associated with them but it does not meet the requirements to be defined as a WCB here. This sentence needs to be revised. This comment also applies to line 445.

We add "do not have a WCB according to the criteria used in this study" in L222, which is of course what we meant. Clearly, with our WCB trajectories we cannot make a statement about shallower and/or slower ascending air. Just for information, note that already Eckhardt et al. (2004) quantified cyclones with and without WCBs (see their Fig. 9) and found a substantial fraction of cyclones without a WCB (according to their latitude dependent WCB criterion).

5. Line 222. Why are the correlations between WCB strength and cyclone deepening rate smaller in the southern hemisphere compared to the northern hemisphere in both ERA-Interim and CESM? This is interesting – is it because SH cyclones are more driven by the low-level baroclinity and less by diabatic processes than in the NH?

We agree that this is interesting, but we can only speculate about the reasons. Your suggestion seems very plausible to us.

6. Line 253, 376, and elsewhere. Units of precipitation. Here in the text it is written mm/h, but in Figure 3 it is mm/6hr. Please check these. Also the precipitation rates (even if mm/6hr) seem to be a little larger than I would expect in a composite as usually on weather maps of day-to-day maximum values of precipitation in one mid-latitude cyclone are around ~5 mm/h. I'd expect the compositing to average / smooth things out leading to smaller values.

Apologies for this typo. It should read "mm $(6 \, h)^{-1}$" as correctly written in Fig. 3.

7. Line 269 and line 433 "remarkably well". This is too strong a statement – it would be more accurate to say there is reasonable agreement. The difference of 0.5 PVU in the intensity of the low-level PV anomaly is, in percentage terms, quite large.

Fine, we changed the formulation to "reasonably well".

8. Line 278. Is the difference in the number of cyclones statistically significant? This could be tested using the number of cyclones each winter and comparing the two populations.

We did not think that these relatively small changes (by 3% and 7%, respectively) are statistically significant. However, in response to the reviewer's question, we performed a t-test. To this aim, we assumed that the 36 winters of ERA-Interim and the 50 winters CESM-HIST/RCP8.5 belong to the same distribution and then checked whether the means significantly differ or not. Whereas the means are not statistically different in the NH, they are in the SH (p-value < 0.01).

9. Line 347. Suggest change "higher" to "larger" as I first though that the WCB peak was moving upwards in the atmosphere.

Thank you, changed as suggested.

10. Line 382–395, Figure 9. Does the vertical tilt change between CESM-HIST and CESM-RCP85? I think this analysis would be a small but valuable addition to this manuscript since some studies (that you cite in the introduction) have shown that the low-level PV anomaly moves downstream which limits the coupling and interaction between the low level and upper level anomaly.

Thanks for this suggestion. We considered the vertical tilt in the composites but could not find an interesting difference between the two climate periods.

Figures and Tables

1. Figure 2. Suggest moving the C1 and C2 labels to a part of the "box" where there is no data as they are hard to see. Also the arrow pointing to the C3 part of the phase space is very easy to miss – can it be made more obvious?

Thank you, the figure has been improved as suggested.

2. Figure 3. Check the units of precipitation (see minor comment #6 above). Add how many cyclones are included in these composites.

The unit mm $(6 \text{ h})^{-1}$ is correct. Number of cyclones has been added above the panels as suggested.

3. Figure 4. When printed, it is very difficult to see the grey shading. Can it be made darker or the edges shown by dashed lines?

Thank you, the figure has been improved. Grey shading in Fig. 4 has been made a bit darker and the edges are shown by dashed lines.

4. Figure 5. This shows the mean number of each type of cyclone per winter. Could some range of the variability be added to this figure (related to minor comment #8)

Thanks, we added the 10th and 90th percentiles of the number of cyclones per winter.

5. Figure 6. Add to the caption what t=0 corresponds to.

Changed as suggested.

6. Figure 7 & 8. Add how many cyclones are included in these composites either to the title (after CESM-HIST etc.) or in the caption.

Information added as suggested.